©c Author(s) 2019. CC BY 4.0 License.



# Kinetic and mechanistic study of the reaction between methane sulphonamide (CH₃S(O)₂NH₂) and OH

Matias Berasategui[1], Damien Amedro[1], Achim Edtbauer[1], Jonathan Williams[1], Jos Lelieveld[1] and John N. Crowley[1]

[1] Division of Atmospheric Chemistry, Max-Planck-Institut für Chemie, 55128 Mainz, Germany

*Correspondence to*: John N. Crowley (john.crowley@mpic.de)

**Abstract.** Methane sulphonamide (MSAM), $CH_3S(O)_2NH_2$, has recently been detected for the first time in ambient air over the Red Sea and the Gulf of Aden where peak mixing ratios of $\approx 60$ pptv were recorded. Prior to this study the rate constant for its reaction with the OH radical and the products thereby formed were unknown, precluding assessment of its role in the atmosphere. We studied the OH-initiated photo-oxidation of MSAM in air (298 K, 700 Torr total pressure) in a photochemical reactor using in situ detection of MSAM and its products by FTIR absorption spectroscopy. The relative rate technique, using three different reference compounds, was used to derive a rate coefficient of $(1.4 \pm 0.3) \times 10^{-13}$ cm$^3$ molecule$^{-1}$ s$^{-1}$. The main end products of the photo-oxidation observed by FTIR were $CO_2$, CO, $SO_2$ and $HNO_3$ with molar yields of $(0.73 \pm 0.11)$, $(0.28 \pm 0.04)$, $(0.96 \pm 0.15)$ and $(0.62 \pm 0.09)$, respectively. $N_2O$ and $HC(O)OH$ were also observed in smaller yields $(0.09 \pm 0.02)$, $(0.03 \pm 0.01)$. Both the low rate coefficient and the products formed are consistent with hydrogen abstraction from the -$CH_3$ group as the dominant initial step. Based on our results MSAM has an atmospheric lifetime with respect to loss by reaction with OH of about 80 days.

## 1 Introduction

Natural emissions of organosulphur compounds from phytoplankton comprise up to 60% of the total sulphur flux into the marine boundary layer (Andreae, 1990; Bates et al., 1992; Spiro et al., 1992) and in remote oceanic areas are the main source of climatically active sulphate aerosols, which can influence the radiation balance at the earth's surface (Charlson et al., 1987; Andreae and Crutzen, 1997). The main organo-sulphur trace gases in the marine boundary layer are dimethyl sulphide ($CH_3SCH_3$, DMS) and its oxidation products dimethyl sulphoxide (DMSO), dimethyl sulphone ($DMSO_2$), methyl sulphonic acid (MSA) and methyl sulphinic acid (MSI) for which atmospheric lifetimes with respect to their degradation by the OH radical vary between hours (DMS) and several weeks ($DMSO_2$).

Recently, the first detection of methane sulphonamide ($CH_3S(O)_2NH_2$, MSAM) in ambient air was made during the Air Quality and Climate Change in the Arabian Basin campaign (AQABA-2017). Mixing ratios of MSAM approached 60 pptv over the Arabian Sea; details of these measurements and a discussion of the likely sources of MSAM in these regions are given in a companion paper (Edtbauer et al., 2019). As MSAM had not been considered to be an atmospheric trace gas prior



to the observations of Edtbauer et al. (2019), there have been no laboratory studies to investigate either its spectroscopy or the kinetics of its reactions with atmospheric radicals, such as OH, so that its atmospheric lifetime and the products formed during its degradation in air were unknown. Crystalline MSAM melts at 363 K, has a boiling point of approximately 453 K and a vapour pressure of < 0.02 Torr at room temperature. Combining carbon, nitrogen, sulphur and oxygen in a single, small molecule, MSAM is an intriguing species not only as an atmospheric trace gas but also from a spectroscopic and

kinetic perspective. Unlike basic alkyl amines such as e.g. $CH_3NH_2$, MSAM contains an acidic $-NH_2$ group (Remko, 2003). This work presents the first kinetic and mechanistic study of the OH induced oxidation of MSAM in air. A reaction mechanism is proposed that, through numerical simulation, describes the time dependent formation of the end-products we observed. From these results, we calculate the lifetime and the likely role of MSAM in the atmosphere.

## 2 Methods

### 2.1 Experimental set-up

The experimental set-up used to study the reaction of OH with MSAM has been described in detail previously (Crowley et al., 1999; Bunkan et al., 2018). Briefly, the reaction vessel was a 44.39 L cylindrical quartz-wall chamber equipped with a White-type multiple reflection mirror system resulting in an 86.3 m optical path length for absorption spectroscopy in the infra-red. The quartz reactor was at room temperature (296 ± 2 K) and for most experiments at 700 Torr of total pressure (1

Torr = 1.333 hPa). Six external, radially mounted, low pressure Hg-lamps emitting mainly at 253.65 nm provided a homogeneous light flux within the reactor for radical generation. A 1000 Torr capacitance manometer was used to measure the pressure inside the reactor.

MSAM and other gases used to generate OH (see below) were mixed in a glass vacuum-line which was connected directly to the reaction chamber by a PTFE piping. Two capacitance manometers (10 Torr and 100 Torr ranges) were used to accurately

measure pressures in the vacuum-line. Owing to its low vapour pressure, MSAM was eluted into the reaction chamber by flowing synthetic air (450 $cm^3$ STP $min^{-1}$, sccm) through a trap containing crystalline MSAM warmed to 333 K, and subsequently through a second cold trap at 298 K (to prevent condensation downstream). This way we could ensure that the saturation vapour pressure of MSAM at 298 K was achieved.

Gas-phase infrared spectra in the range of 4000–600 $cm^{-1}$ were recorded with a resolution of 2 $cm^{-1}$ from 16 co-added

interferograms (128 scans for the background) using a Fourier Transform Infra-Red (FTIR) spectrometer (Bruker Vector 22) equipped with an external photoconductive mercury-cadmium-telluride (MCT) detector cooled to liquid nitrogen temperature. OPUS-software was used to analyze and manipulate the IR spectra. Interferograms were phase-corrected (Mertz) and Boxcar apodized with a zero-filling-factor of 4. Most of the products obtained ($CO_2$, CO, HC(O)OH, $HNO_3$ and $SO_2$) were identified and quantified from the IR reference spectra of pure samples under similar experimental conditions

(700 Torr and 298.2 K, Figure S1).



The low vapour pressure of MSAM precluded accurate dosing into the chamber and thus generation of a calibration spectrum. In order to calibrate the infra-red absorption features of MSAM we oxidized it in air and then conducted a sulphur and nitrogen balance of the products. As discussed below, the only sulphur-containing product detected from MSAM degradation was $SO_2$ (which can easily be calibrated) and the only nitrogen contained products were $HNO_3$ and $N_2O$, which can also be calibrated. Experiments in which MSAM was almost completely converted to known amounts of $SO_2$, $HNO_3$ and $N_2O$ thus provided an indirect calibration (via assumption of 100% sulphur or nitrogen balance) of its concentration and thus IR-cross-sections.

## 2.2 Generation of OH

OH was generated by the 254 nm photolysis of $O_3$ in the presence of $H_2$.

$$O_3 + h\nu \ (254 \ nm) \quad \rightarrow \quad O(^1D) + O_2 \quad\quad\quad\quad (R1)$$

$$O(^1D) + H_2 \quad \rightarrow \quad OH + H \quad\quad\quad\quad (R2)$$

$$H + O_2 + M \quad \rightarrow \quad HO_2 + M \quad\quad\quad\quad (R3)$$

Further reactions that cycle OH and $HO_2$ (e.g. $OH + H_2$, $H + O_3$, $HO_2 + O_3$) are listed in Table S1 of the supplementary information.

In a typical experiment, the starting concentrations of $O_3$ and $H_2$ were $\approx 5 \times 10^{14}$ and $\approx 5 \times 10^{16}$ molecule $cm^{-3}$. The large excess of $H_2$ ensures that $O(^1D)$ does not react with MSAM. As described previously (Bunkan et al., 2018), this scheme generates not only OH radicals but (via e.g. (R3) also $HO_2$. $HO_2$ is not expected to react with MSAM but will influence the course of secondary reactions in this system (e.g. by reacting with organic peroxy radicals) and thus the end-product distribution, as described in detail in section 3.5. Simulations of the radical concentrations when generating OH in this manner indicate that the $HO_2$ / OH ratio is approximately 30, with individual concentrations of $\approx 1 \times 10^{11}$ molecule $cm^{-3}$ $HO_2$ and $3 \times 10^9$ molecule $cm^{-3}$ OH.

As OH source, the photolysis of $O_3$ in the presence of $H_2$ has the advantage over other photochemical sources (e.g. photolysis of $H_2O_2$, HONO or $CH_3ONO$) that neither $H_2$ nor $O_3$ have strong absorption features in the infra-red, resulting in a less cluttered spectrum which simplifies retrieval of concentration-time profiles of reactants and products.

## 2.3 Chemicals

A commercially available sample of methane sulphonamide (Alfa Aesar, > 98%) was used. $O_3$ was generated by flowing synthetic air (Westfalen) through a stainless steel tube that housed a low-pressure Hg-lamp (PenRay) emitting at 184.95 nm. Synthetic air (Westfalen, 99.999%), $H_2$ (Westfalen, 99.999%), $CO_2$ (Westfalen 99.995%), CO (Westfalen, 99.997 %), $SO_2$ (Air Liquide, 1 ppmv in air) and HC(O)OH (Sigma Aldrich) were obtained commercially. Anhydrous nitric acid was prepared by mixing $KNO_3$ (Sigma Aldrich, 99%) and $H_2SO_4$ (Roth, 98%), and condensing $HNO_3$ vapour into a liquid nitrogen trap.



## 2.4 Relative rate constant determination

The rate constant ($k_4$) of the reaction between OH and $CH_3SO_2NH_2$ (Reaction R4) was measured using the relative rate method using (in different experiments) formic acid (HC(O)OH), acetone ($CH_3C(O)CH_3$) and methanol ($CH_3OH$) as reference compounds.

| | | | |
|---|---|---|---|
| $CH_3S(O)_2NH_2 + OH$ | → | Products | (R4) |
| $CH_3C(O)CH_3 + OH$ | → | Products | (R5) |
| $HC(O)OH + OH$ | → | Products | (R6) |
| $CH_3OH + OH$ | → | Products | (R7) |

Relative rate constants were derived by monitoring the depletion of one or more IR-features of MSAM relative to those of the reference compounds. The following expression links the depletion factors (e.g. $\ln([MSAM]_0/[MSAM]_t)$) to the relative rate coefficient:

$$\ln\left(\frac{[MSAM]_0}{[MSAM]_t}\right) = \frac{k_4}{k_{ref}}\ln\left(\frac{[ref]_0}{[ref]_t}\right) \tag{1}$$

where $[MSAM]_0$, $[MSAM]_t$, $[ref]_0$, and $[ref]_t$ are the concentrations of MSAM and reference compound at times 0 and t; $k_4$ and $k_{ref}$ are the rate constants for reactions of OH with the MSAM and reference, respectively. As only relative changes in the IR-signal are used in the analysis, absolute concentrations are not required as long as the absorption features used display a linear relation with concentration. Plots of $\ln([MSAM]_0 / [MSAM]_t)$ versus $\ln([ref]_0 / [ref]_t)$ should therefore be linear, pass through the origin, and have a slope of $k_4 / k_{ref}$. The experimental procedure consisted of filling the cell with a mixture of MSAM / reference / $O_3$ / $H_2$ / $N_2$ and allowing it to mix (5-10 min) prior to subjecting the mixture to 253.65 nm radiation whilst monitoring IR features at 3-5 min intervals.

This analysis inherently assumes that the only loss process for MSAM and the reference molecules is reaction with OH. Experiments in which the starting gas-mixture was allowed to stand for several hours with no discernible loss of MSAM, formic acid, methanol or acetone confirmed that none of these gases are lost to the wall or react with $O_3$ to a significant extent. From observation of MSAM and $O_3$ mixtures we were able to derive an upper limit for the reaction of MSAM with $O_3$ of $1 \times 10^{-19}$ $cm^3$ $molecule^{-1}$ $s^{-1}$.

## 3 Results and Discussion

### 3.1 Vibrational characterisation of $CH_3SO_2NH_2$

The experimental, infra-red absorption spectrum of MSAM (Fig. 1) shows characteristic bands corresponding to $SO_2$ stretching vibrations at 1385 and 1172 $cm^{-1}$, the $NH_2$ wagging vibration at 857 $cm^{-1}$, stretching vibrations at 3476 and 3380 $cm^{-1}$, bending at 1551 $cm^{-1}$, and the $CH_3$ wagging band at 976 $cm^{-1}$. Assignment of the infra-red features (Table 1) was made by comparison with a theoretical spectrum calculated using density functional theory (DFT) at the B3LYP/6-311++G(3d,2p) and B3LYP/aug-cc-pVTZ-pp levels of theory for the vibrational characterization of MSAM in gas phase. These basis sets



should be sufficient to describe the relative energies for the isomers. Harmonic vibrational frequencies and zero-point energies (ZPE) were calculated at these levels of theory to check whether the stationary points obtained were either isomers or first-order transition states (all calculated conformers had only real frequencies). The high accuracy energy method Gaussian-4 with Møller-Plesset expansion truncated at second order (G4MP2) was also used for the calculation of the barrier energies. The determination of the Hessian matrix also enabled the calculation of the thermochemical quantities for the conformers at 298.15 K. All symmetry restrictions were turned off in the calculations. All calculations were run with the Gaussian 09 program package (Frisch, 2010). Assuming that the point group for the molecule is Cs, all 24 fundamental modes should be both IR and Raman active, fourteen of them belonging to A′ representation and ten to A″. All the vibrational frequencies are real and positive. The assignments in Table 1 were made from an evaluation of the normal modes displacement vectors; as many of the modes are strongly coupled, this information is rather subjective. The frequencies of the absorption bands of the theoretical spectrum displayed in Fig.1 were adjusted by a scaling factor of 0.968 ± 0.019 recommended for the B3LYP/aug-cc-pVTZ level of theory (column "ratio" in Table 1) (Halls et al., 2001).

If the "cold trap" is removed, extra absorption bands originating from the MSAM-dimer are observed. These slowly disappear with time as the condensation of the low-volatility dimer to the reactor surfaces takes place. Fig. S1 of the supplementary information shows the IR spectra of the dimer after the subtraction of the monomer. A complete characterization of the vibrational modes is presented in Table S1 of the supplementary information. According to our calculations, two hydrogen-bond interactions between the -HNH···OSO- are formed in the dimer which produce a bathochromic shift of the absorption bands. For each kinetic experiment we ensure that no dimer band is present in the initial spectrum.

**3.2 Relative rate measurements: Determination of $k_4$**

Once the concentrations of MSAM and the reference compound were stable (i.e. mixing in the chamber was complete) the Hg-lamps were switched on for a period of typically one hour during which FTIR spectra (duration of ~20 seconds) were obtained every few minutes. The concentrations of the reactants in each individual experiment can be found in Table 2. Figure 2 shows the loss of absorption features due to MSAM and the reference compound (in this case acetone) at different times during the experiment.

The depletion of MSAM was quantified by integrating the Q-branch of the 857, 1172, 1383 cm$^{-1}$ absorption bands, and the complete absorption band at 3380 cm$^{-1}$. The relative depletion of the MSAM absorption-features agreed to within ~ 5%. Depletion of the reference gases were quantified by integrating their absorption bands at 1221-1249 cm$^{-1}$ (acetone), 2788-3070 cm$^{-1}$ (methanol) and 1073-1133 cm$^{-1}$ (formic acid). An alternative analysis procedure, in which the relative depletion of MSAM was derived by scaling a reference spectrum of MSAM (e.g. that obtained prior to photolysis) to match those at various times after photolysis was also used. The depletion factors thus obtained were indistinguishable from those using individual absorption features.





Figure 3 shows plots of the depletion factors for MSAM versus those of the three reference compounds following exposure to OH radicals in 700 Torr of synthetic air at 296 K. A linear least-squares analysis of the data gives rate constant ratios $k_4 / k_5 = (0.778 \pm 0.008)$, $k_4 / k_6 = (0.307 \pm 0.004)$ and $k_4 / k_7 = (0.158 \pm 0.002)$ where the quoted errors are two standard deviations. Table 2 summarizes the experimental conditions and the rate coefficient ratios obtained when using each MSAM absorption band. The difference between the rate coefficient ratios obtained for the three absorption bands experiments is

always less than 5%. The rate constant ratios were placed on an absolute basis using evaluated rate coefficients (Atkinson et al., 2006; IUPAC, 2019) whereby $k_5 = (1.8 \pm 0.36) \times 10^{-13}$, $k_6 = (4.5 \pm 1.8) \times 10^{-13}$, and $k_7 = (9.0 \pm 1.8) \times 10^{-13}$ cm$^3$ molecule$^{-1}$ s$^{-1}$. We derive values of $k_4$ (relative to acetone) = $(1.40 \pm 0.28) \times 10^{-13}$, $k_4$ (relative to formic acid) = $(1.38 \pm 0.55) \times 10^{-13}$, and $k_4$ (relative to methanol) = $(1.42 \pm 0.28) \times 10^{-13}$ cm$^3$ molecule$^{-1}$ s$^{-1}$ (where the uncertainties include uncertainty associated with the evaluated rate coefficients for $k_5$, $k_6$ and $k_7$). The values of $k_4$ obtained using the three deferent reference compounds

are, within experimental uncertainties, identical, indicating the absence of significant systematic errors associated with the use of the reference reactants. We prefer the value of $k_4$ from the experiment using acetone as reference. For acetone, the relative rate constant is close to unity and the rate coefficient for OH has been extensively studied and is associated with low uncertainty. The preferred value of the rate coefficient, $k_4$, is $(1.4 \pm 0.3) \times 10^{-13}$ cm$^3$ molecule$^{-1}$ s$^{-1}$ where the uncertainty is $2\sigma$.

### 3.3 Product yields

In order to identify and quantify the end-products of the title reaction in air, approximately $(6.25 \pm 0.75) \times 10^{13}$ molecule cm$^{-3}$ of MSAM, $4.04 \times 10^{14}$ molecule cm$^{-3}$ of $O_3$ and $1.00 \times 10^{15}$ molecule cm$^{-3}$ of $H_2$ were loaded into the chamber at a total pressure of 700 Torr of synthetic air and 298 K. Subsequent to initiation of the reaction between OH and MSAM by switching the Hg-lamps on, IR-spectra (700 – 4000 cm$^{-1}$) were taken at 300 seconds intervals.

Figure 4 shows the initial spectrum of the gas mixture with $O_3$ bands at 903-1068 and 2064-2134 cm$^{-1}$(A), the spectrum after 3000 s (B) showing depletion of MSAM and formation of products and the final products after the disappearance of the Q-branches (at 1172 and 1385 cm$^{-1}$) of MSAM after 5700 s (C). The IR-absorption bands of water vapour have been subtracted from the spectra. Both $CO_2$ (2387-2300 cm$^{-1}$) and CO (2226-2050 cm$^{-1}$) are observed from the photolysis of compounds adsorbed on the walls and surfaces of the cell, and do not result solely from MSAM degradation.

Figure 4 also displays reference spectra (measured at the same temperature and pressure) of the compounds we identified as reaction products. Other than $CO_2$ and CO, nitric acid ($HNO_3$) and sulphur dioxide ($SO_2$) are easily identified, with weak features from $N_2O$, $NO_2$ and formic acid (HC(O)OH) also apparent. The absorption of each product was converted to a concentration using calibration curves that were obtained at the same pressure and temperature (see Fig. S2 of the supplementary information).

Figure 5 plots the concentration of $SO_2$ (the only sulphur containing product observed), the sum of $HNO_3 + N_2O$ (the total reactive nitrogen observed as product) and the sum of $CO_2 + CO + HC(O)OH$ (total carbon containing products observed)





versus the fractional depletion of MSAM. The concentrations after 6000 seconds (when ~90% of the MSAM had reacted) were: $[SO_2] = 5.74 \times 10^{13}$, $[HNO_3] = 3.15 \times 10^{13}$, $[N_2O] = 4.17 \times 10^{12}$, $[CO_2] = 4.13 \times 10^{13}$, $[CO] = 1.65 \times 10^{13}$ and $[HC(O)OH] = 1.77 \times 10^{12}$ molecules $cm^{-3}$. MSAM contains one atom each of sulphur, nitrogen and carbon. If $SO_2$, reactive

nitrogen and carbon are conserved, we can derive initial concentrations of MSAM (from the slope) of $6.12 \pm 0.08 \times 10^{12}$ molecule $cm^{-3}$ (based on the sulphur balance), $5.10 \pm 0.05 \times 10^{12}$ molecule $cm^{-3}$ (based on the nitrogen balance) and $7.4 \pm 0.2 \times 10^{12}$ molecule $cm^{-3}$ (based on the carbon balance at the maximum fractional depletion of MSAM). As already mentioned, total carbon is very likely to be overestimated due to its formation and desorption at/from the walls of the chamber. As the main nitrogen product is $HNO_3$, which has a large affinity for surfaces and which is likely to be lost to the walls, we also

expect that use of reactive nitrogen will result in an underestimation of the initial MSAM concentration. For these reasons we consider that the best method to estimate the initial concentration of MSAM is via the formation of $SO_2$. Figure S3 of the supplementary information illustrates the strict proportionality between the relative change of the $SO_2$ and MSAM absorption features.

From this experiment we derive an initial MSAM concentration of $(6.1 \pm 1.0) \times 10^{13}$ molecules $cm^{-3}$ and use this value to

derive the absorption cross-sections for MSAM (these are given in Fig. 1), which can be used to calculate initial concentrations in all other experiments. Figure 6a presents a plot of $\Delta[product]$ versus $-\Delta[MSAM]$ for the same experiment. Apart from CO, we observe a roughly linear relationship for all products. Time dependent yields of each product are displayed in Fig. 6b. Whereas the yields of $SO_2$, $CO_2$, $N_2O$ and $HC(O)OH$ are, within experimental scatter roughly constant, that of $HNO_3$ (black line) reaches a constant value only after 800 seconds, indicating that it is not formed directly but in a

secondary reaction. In contrast, the CO yield is initially larger than unity (indicative of extra sources from the chamber walls) and then decreases with time. The time dependence of the CO-yield indicates that it is removed (via reaction with OH) to form $CO_2$.

The molar yields (after 6000 seconds of photolysis) of the products obtained at 298 K and 700 Torr of synthetic air are: $\Phi(SO_2) = 0.96 \pm 0.15$, $\Phi(HNO_3) = 0.62 \pm 0.09$, $\Phi(N_2O) = 0.09 \pm 0.02$, $\Phi(CO) = 0.28 \pm 0.04$, $\Phi(CO_2) = 0.73 \pm 0.11$,

$\Phi(HC(O)OH) = 0.03 \pm 0.01$. The slight deviation of $\Phi(SO_2)$ from unity stems from the fact that the quoted yield is at a fixed time, whereas the initial MSAM concentration was derived using all the $SO_2$ data in this experiment as described above. As $N_2O$ contains two N-atoms, the nitrogen balance is thus $0.80 \pm 0.13$. It is likely that some $HNO_3$ is lost to reactor surfaces, explaining the deviation from unity. Note that if we had used the nitrogen balance to derive the MSAM IR-cross-sections, the $SO_2$ yield would have exceeded unity.

**3.4 Reaction mechanism**

The time dependent formation of $HNO_3$, $SO_2$, $N_2O$ and CO provide important clues to the reaction mechanism. Addition to the S-atom is not possible so that the initial step will be abstraction of hydrogen by the OH radical, either from the $-CH_3$ group (Reaction R8a) or from the $-NH_2$ group (Reaction R8b):





| | | | |
|---|---|---|---|
| $CH_3SO_2NH_2 + OH$ | $\rightarrow$ | $CH_2SO_2NH_2 + H_2O$ | (R8a) |


| | | | |
|---|---|---|---|
| $CH_3SO_2NH_2 + OH$ | $\rightarrow$ | $CH_3SO_2NH + H_2O$ | (R8b) |

Based on results of previous studies of the reactions of OH with trace-gases containing both $CH_3$ and $-NH_2$ entities (e.g. $CH_3NH_2$ or $CH_3C(O)NH_2$) we expect abstraction at the $-CH_3$ group (Reaction R8a) to dominate (Onel et al., 2014; Borduas et al., 2015; Butkovskaya and Setser, 2016). H-abstraction at the methyl-group is also consistent with a rate coefficient for R4 that is very similar to that for OH + acetone.

### 3.4.1 Abstraction from the $-CH_3$ group

In section 3.4.1 we focus on the fate of the peroxy radical, $OOCH_2SO_2NH_2$, formed by reaction of initially formed $CH_2SO_2NH_2$ with $O_2$ (R9). The most important reactions of organic peroxy radicals are self-reactions (R10) or reactions with NO (R11), $NO_2$ (R12), or $HO_2$ (R13).

| | | | |
|---|---|---|---|
| $CH_2SO_2NH_2 + O_2$ | $\rightarrow$ | $OOCH_2SO_2NH_2$ | (R9) |
| 230   $2\ OOCH_2SO_2NH_2$ | $\rightarrow$ | $OCH_2SO_2NH_2 + O_2$ | (R10) |
| $OOCH_2SO_2NH_2 + NO$ | $\rightarrow$ | $OCH_2SO_2NH_2 + NO_2$ | (R11) |
| $OOCH_2SO_2NH_2 + NO_2$ | $\rightarrow$ | $O_2NOOCH_2SO_2NH_2$ | (R12) |
| $OOCH_2SO_2NH_2 + HO_2$ | $\rightarrow$ | $HOOCH_2SO_2NH_2 + O_2$ | (R13) |

Peroxy nitrates such as the one formed in Reaction (R12) are thermally unstable with respect to dissociation back to

reactants at room temperature and given the very low concentrations of $NO_2$ in our system, Reaction (R12) will not play a significant role in this study.

The oxy-radical, $OCH_2SO_2NH_2$ formed in Reactions (R10) and (R11) will react with $O_2$ to produce an aldehyde (Reaction R14). Alternatively, it could undergo C-S bond cleavage (Reaction R15) to form formaldehyde ($CH_2O$) and the $SO_2NH_2$ radical:

| | | | |
|---|---|---|---|
| 240   $OCH_2SO_2NH_2 + O_2$ | $\rightarrow$ | $HC(O)SO_2NH_2 + HO_2$ | (R14) |
| $OCH_2SO_2NH_2$ | $\rightarrow$ | $CH_2O + SO_2NH_2$ | (R15) |

The fate of $HC(O)SO_2NH_2$ will be reaction with OH to form $C(O)SO_2NH_2$ (R16) which will dissociate to form CO + $SO_2NH_2$ (R17). The rate coefficient for reaction (R16) is expected to be $\approx 10^{-11}$ $cm^3$ $molecule^{-1}$ $s^{-1}$ as for many similar reactions of OH with aldehydes (e.g. $CH_3CHO$).

| | | | |
|---|---|---|---|
| 245   $HC(O)SO_2NH_2 + OH$ | $\rightarrow$ | $C(O)SO_2NH_2 + H_2O$ | (R16) |
| $C(O)SO_2NH_2$ | $\rightarrow$ | $SO_2NH_2 + CO$ | (R17) |

The predominant fate of formaldehyde will be reaction with OH to form CO and subsequently $CO_2$:

| | | | |
|---|---|---|---|
| $CH_2O + OH$ | $\rightarrow$ | $HCO + H_2O$ | (R18) |
| $HCO + O_2$ | $\rightarrow$ | $HO_2 + CO$ | (R19) |
| 250   $CO + OH\ (+O_2)$ | $\rightarrow$ | $CO_2 + HO_2$ | (R20) |

But it may also react with $HO_2$ to form formic acid:



| $CH_2O + HO_2 + M$ | $\rightarrow$ | $HOCH_2OO + M$ | (R21) |
| $2\ HOCH_2OO$ | $\rightarrow$ | $HOCH_2O + O_2$ | (R22) |
| $HOCH_2O + O_2$ | $\rightarrow$ | $HC(O)OH + HO_2$ | (R23) |

The above reactions explain, at least qualitatively, the observed formation of CO, $CO_2$ and HC(O)OH. Note that the room temperature rate coefficient for reaction of OH with HCHO is large ($8.5 \times 10^{-12}$ $cm^3$ molecule$^{-1}$ s$^{-1}$, Atkinson et al. (2006)) compared to that for reaction with CO ($2.2 \times 10^{-13}$ $cm^3$ molecule$^{-1}$ s$^{-1}$ Atkinson et al. (2006)), which explains why CO was observed as an intermediate product at high concentrations whereas HCHO was not.

The likely fate of the $SO_2NH_2$ radical formed in Reaction (R15) is either reaction with $O_2$ to generate $SO_2NH$ or dissociation
by S-N bond-scission to produce $SO_2$ and the $NH_2$ radical.

| $SO_2NH_2 + O_2$ | $\rightarrow$ | $SO_2NH + HO_2$ | (R24) |
| $SO_2NH_2$ | $\rightarrow$ | $SO_2 + NH_2$ | (R25) |

We did not observe features in the IR- spectrum that that could be assigned to $SO_2NH$ based on the spectrum reported by Deng et al. (2016) and propose that reaction (R25) is the source of $SO_2$ as a major reaction product. By analogy with the
thermal decomposition of the similar $CH_3SO_2$ radical, which dissociates to $CH_3$ and $SO_2$ on a millisecond time scale (Ray et al., 1996), we expect $SO_2NH_2$ to decompose stoichiometrically to $SO_2$ and $NH_2$ on the time scale of our experiments. The $NH_2$ radical, is known to react with $O_3$, $HO_2$ and $NO_2$ (IUPAC, 2019):

| $NH_2 + O_3$ | $\rightarrow$ | $NH_2O + O_2$ | (R26) |
| $NH_2 + HO_2$ | $\rightarrow$ | $NH_2O + OH$ | (R27a) |
| 270 $NH_2 + HO_2$ | $\rightarrow$ | $HNO + H_2O$ | (R27b) |
| $NH_2 + NO_2$ | $\rightarrow$ | $N_2O + H_2O$ | (R28) |

$NH_2O$ rearranges within ~1 ms to NHOH (Kohlmann and Poppe, 1999), which then reacts with OH or $O_2$ to generate HNO:

| $NHOH + OH$ | $\rightarrow$ | $HNO + H_2O$ | (R29) |
| $NHOH + O_2$ | $\rightarrow$ | $HNO + HO_2$ | (R30) |

The fate of HNO is the reaction with OH or $O_2$ to generate NO (reactions R31 and R32):

| $HNO + O_2$ | $\rightarrow$ | $NO + HO_2$ | (R31) |
| $HNO + OH$ | $\rightarrow$ | $NO + H_2O$ | (R32) |

High concentrations of $O_3$ ($\approx 10^{14}$ molecule cm$^{-3}$) and $HO_2$ ($\approx 10^{11}$ molecule cm$^{-3}$) in our system ensure that NO is converted to $NO_2$ in less than 1s, explaining the non-observation of the IR absorption features of NO.

| 280 $NO + O_3$ | $\rightarrow$ | $NO_2 + O_2$ | (R33) |
| $NO + HO_2$ | $\rightarrow$ | $NO_2 + OH$ | (R34) |

Finally, $NO_2$ in this system will react with OH to form the main reactive nitrogen compound we observed, $HNO_3$.

| $NO_2 + OH + M$ | $\rightarrow$ | $HNO_3 + M$ | (R35) |

Thus far we have not considered the possible, competitive reaction of the peroxy radical, $OOCH_2SO_2NH_2$ with $HO_2$
(Reaction R13), which is expected to result in the formation of a peroxide, $HOOCH_2SO_2NH_2$. The most likely fate of


$HOOCH_2SO_2NH_2$ is reaction with OH for which (via comparison with $CH_3OOH$) a rate coefficient close to $1\text{-}5 \times 10^{-12}$ $cm^3$ molecule$^{-1}$ s$^{-1}$ may be expected with H-abstraction from both the peroxide group (Reaction R36) or the adjacent carbon (Reaction R37).

$$HOOCH_2SO_2NH_2 + OH \quad \rightarrow \quad OOCH_2SO_2NH_2 + H_2O \tag{R36}$$

$$HOOCH_2SO_2NH_2 + OH \quad \rightarrow \quad HOOCHSO_2NH_2 + H_2O \tag{R37}$$

Reaction (R36) regenerates the peroxy radical, whereas the $HOOCHSO_2NH_2$ radical may decompose (Reaction R38) to form formic acid $HC(O)OH$ or via Reaction (R39) to form the same aldehyde that is generated in Reaction (R14), whilst regenerating OH:

$$HOOCHSO_2NH_2 \quad \rightarrow \quad HC(O)OH + SO_2NH_2 \tag{R38}$$

$$HOOCHSO_2NH_2 \quad \rightarrow \quad OH + HC(O)SO_2NH_2 \tag{R39}$$

The final products are thus the same as those resulting from the self-reaction of the peroxy radical. The path from MSAM to the observed end-products including the reactive intermediates that were not observed is illustrated in Fig. 7.

### 3.4.2 Abstraction from the –NH₂ group

In analogy to the reaction between $CH_3C(O)NH_2$ and OH (Barnes et al., 2010), H- abstraction from the -NH₂ group is 300 expected to result in decomposition of the initially formed $CH_3SO_2NH$ radical via C-S bond fission.

$$CH_3SO_2NH \quad \rightarrow \quad CH_3 + SO_2NH \tag{R40}$$

The methyl radical would react with $O_2$ to form the methyl-peroxy radical and in subsequent reactions (via $CH_3O$) would result in $CH_2O$ formation. As discussed above $CH_2O$ will be efficiently oxidized to CO and $CO_2$ in this system. However, the characteristic IR-absorption bands (Deng et al., 2016) of the $SO_2NH$ product were not observed in our experiments and 305 calculations at the G4MP2 level of theory indicate that Reaction (R40) is endothermic (by 137 kJ mol$^{-1}$). We conclude that H-abstraction from the –NH₂ group is a minor channel.

### 3.4 Kinetic Simulation

The proposed reaction mechanism (considering initial reaction by H-abstraction from the –$CH_3$ group only) was tested by kinetic simulation using the KINTECUS program package (Ianni, 2015). The reactions used in the chemical scheme and the 310 associated rate coefficients are presented in Table S2. Where experimental rate coefficients were not available, we used rate parameters from similar reactions, and rationalize these choices in the text associated with Table S2.

Figure 8 shows the variation of the concentrations of the reagent, intermediates and products observed as a function of time in an experiment conducted at 298 K and 700 Torr of synthetic air. The plotted uncertainties result from uncertainty associated with the absorption cross-sections (5 - 12%) and uncertainty in deriving the areas of the absorption bands areas 315 (less than 3% in all cases). For MSAM, an uncertainty of $\approx 25\%$ is expected, based on the indirect method of calibration (see Section 3.2).





The experimental and simulated concentration-time profiles are in good agreement except for $CO_2$. As described in section 3.5, $CO_2$ is generated from the cell walls and cannot be used quantitatively. The good agreement with the $N_2O$ (formed from $NH_2$ in Reaction R28) and $HNO_3$ experimental data suggests that the fate of $NH_2$ (the only source of reactive nitrogen in this

system) is accurately described in the model. Note that the wall loss rate of $HNO_3$ ($1 \times 10^{-5}$ s$^{-1}$) in the simulation was adjusted to match the $HNO_3$ profile. The simulated amount of $HNO_3$ lost to the wall at the end of the experiment was ≈ 14% of that formed, which helps to explain the non-unity yield of gas-phase nitrogen compounds.

The grey line in Fig. 8 represents the sum of $SO_2 + SO_3 + H_2SO_4$, i.e. all model trace gases containing sulphur, which, in the absence of IR absorption features of $SO_3$ or $H_2SO_4$, we equate to $SO_2$. We now draw attention to the fact that $SO_2$ (the yield

of which is constant with time, see Fig. 6) is only well simulated if we neglect its removal by OH (Reaction R41).

$$OH + SO_2 + M \quad\rightarrow\quad HOSO_2 + M \tag{R41}$$

Otherwise, using the preferred rate constant (IUPAC, 2019) at 700 Torr and 298 K of $9.0 \times 10^{-13}$ cm$^3$ molecule$^{-1}$ s$^{-1}$ we find that the simulated $SO_2$ concentration is significantly reduced and its yield is time dependent. At one bar of air, collisionally stabilized $HOSO_2$ is converted within 1 μs to $HO_2$ and $SO_3$. In the atmosphere, $SO_3$ reacts with $H_2O$ to form $H_2SO_4$ (R43).

The conversion of $SO_3$ to $H_2SO_4$ may be suppressed under our "dry" conditions.

$$HOSO_2 + O_2 \quad\rightarrow\quad HO_2 + SO_3 \tag{R42}$$

$$SO_3 + (H_2O)_n \quad\rightarrow\quad H_2SO_4 + (H_2O)_{n-1} \tag{R43}$$

$SO_2$ should therefore not behave like a stable end-product in our experiments, but be converted to more oxidized forms. In order to confirm that $SO_2$ is a stable end product in our experiments, we measured the relative rate of loss of $SO_2$ and acetone

under the same experimental conditions (Fig. S4 of the supplementary information). The apparent, relative rate constant $k_{41} / k_5$ was 0.46, which converts to an effective rate constant for $SO_2$ loss of $8.2 \times 10^{-14}$ cm$^3$ molecule$^{-1}$ s$^{-1}$. This is more than a factor of ten lower than the preferred value, indicating that the net rate of the OH-induced $SO_2$ loss in our system is much lower than expected and not simply governed by the rate constant for the forward reaction to form $HOSO_2$. The reformation of $SO_2$ under our experimental conditions is subject of ongoing experiments in this laboratory, which are beyond the scope

of the present study. We note that the unexpected behaviour of $SO_2$ does not significantly impact on the conclusions drawn from the present study.

### 3.5 Atmospheric Implications

The rate coefficient for a number of tropospheric, organo-sulphur trace gases are listed in Table 3. The rate coefficient for the title reaction is significantly lower than those for $CH_3SCH_3$ ($CH_3SCH_3$) and $CH_3S(O)CH_3$, (DMSO) for which reaction

with OH is the major atmospheric loss process (lifetimes of hours), but comparable to $CH_3S(O)_2CH_3$ which also has two S=O double bonds. However, as for most tropospheric trace gases, the lifetime of MSAM will be controlled by a number of processes including photolysis, reactions with the three major oxidants, OH, $NO_3$ and $O_3$ as well as dry deposition ($k_{dd}$) and heterogeneous uptake to particles ($k_{het}$), followed by wet deposition.





The lack of C=C double-bonds in MSAM suggest that the reaction with $O_3$ will be a negligible sink, which is confirmed by

the low upper limit to the rate constant of $1 \times 10^{-19}$ cm$^3$ molecule$^{-1}$ s$^{-1}$ described in section 2.4. Whereas the reaction with $NO_3$ represents an important loss mechanism for DMS, we do not expect this to be important for MSAM. $CH_3SCH_3$ reacts with $NO_3$ (despite lack of a C=C double bond) as the high-electron density around the sulphur atom enables a pre-reaction complex to form prior to H-abstraction. This mechanism is not available for MSAM because the electron density around the sulphur atom is reduced by the two oxygen atoms attached to it, which also provide steric-hindrance.

Owing to its low vapour pressure, we were unable to measure the UV-absorption spectrum of MSAM, but note that it was not photolysed at a measureable rate by the 254 nm radiation in our study. We conclude that photolysis in the troposphere, where actinic flux only at wavelengths above $\geq$ 320 nm is available is a negligible sink of MSAM.

Therefore, the lifetime of MSAM can be approximated by:

$$\tau_{MSAM} = \frac{1}{k_4[OH] + k_{dd} + k_{het}} \qquad (2)$$

Using our overall rate coefficient, $k_4 = 1.4 \times 10^{-13}$ cm$^3$ molecule$^{-1}$ s$^{-1}$ for the title reaction and taking a diel-averaged OH concentration of $1 \times 10^6$ molecules cm$^{-3}$, we can use equation (2) to calculate a first-order loss rate constant of $k_4[OH] = 1.4 \times 10^{-7}$ s$^{-1}$. Which is equivalent to a lifetime of $\approx$ 80 days.

MSAM is highly soluble and a dry deposition velocity of $\approx$ 1 cm s$^{-1}$ to the ocean has been estimated (Edtbauer et al., 2019). Combined with a marine boundary height of $\approx$ 750 $\pm$ 250 m, this results in a loss rate coefficient of $1.3 \times 10^{-5}$ s$^{-1}$ or a lifetime

with respect to uptake to the ocean of less than one day. Wet deposition is also likely to play a role, which may limit the MSAM lifetime to days under rainy conditions to weeks in dry regions.

To a first approximation the heterogeneous loss rate of a trace gas to a particle is given by:

$$k_{het} = 0.25 \, \gamma \bar{c} A \qquad (3)$$

where $\gamma$ is the uptake coefficient which represents the net efficiency (on a per collision basis) of transfer of MSAM from the

gas-phase to the particle phase), $\bar{c}$ is the mean molecular velocity of MSAM (~26000 cm s$^{-1}$) and $A$ is the surface area density of particles (in cm$^2$ cm$^{-3}$) for which a typical value in low to moderately polluted regions would be $1 \times 10^{-6}$ cm$^2$ cm$^{-3}$. A rather low uptake coefficient of ~$2 \times 10^{-5}$ would then be sufficient to compete with MSAM loss due to reaction with OH, but a value of $2 \times 10^{-3}$ would be necessary to compete with dry-deposition.

## 4 Conclusions

The rate coefficient for reaction of methane sulphonamide (MSAM) with OH was determined using the relative rate method as $(1.4 \pm 0.3) \times 10^{-13}$ cm$^3$ molecule$^{-1}$ s$^{-}$. The major, stable, quantifiable sulphur and nitrogen containing end-products of the reaction are $SO_2$ and $HNO_3$ with molar yields of $(0.96 \pm 0.15)$ and $(0.62 \pm 0.09)$, respectively. CO and $CO_2$ are the dominant carbon-containing products. $N_2O$ and HC(O)OH were also observed at lower yields of $(0.09 \pm 0.02)$ and $(0.03 \pm 0.01)$, respectively. The end-products (and the low rate coefficient) are consistent with an initial abstraction by OH from the $CH_3$



group. Based on our results MSAM has an atmospheric lifetime with respect to loss by reaction with OH of about 80 days, indicating that other processes (e.g. deposition) will likely dominate.





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





**Table 1.** Experimental and Calculated Vibrational Wavenumbers for MSAM.

| Mode-symm. | mode | Frequencies (cm⁻¹) | | | | Mode description |
|---|---|---|---|---|---|---|
| | | Experiment [a,b] | 6-31++(d,p) [a,b] | Aug-CC-pVTZ [a,b] | ratio | |
| A'' | $\nu_1$ | 3476 (18.8) | 3627 (18.8) | 3612 (19.6) | 0.958 | $NH_2$ asym. stretch |
| | $\nu_3$ | | 3193 (<0.1) | 3169 (0.2) | | $CH_3$ deformation |
| | $\nu_7$ | | 1461 (1.1) | 1457 (0.4) | | $CH_3$ rocking |
| | $\nu_{10}$ | 1383 (100) | 1322 (100) | 1342 (100) | 1.048 | $SO_2$ asym. stretch |
| | $\nu_{12}$ | | 1085 (2.2) | 1087 (1.5) | | $NH_2$ rocking |
| | $\nu_{14}$ | | 981 (0.3) | 972 (0.3) | | $CH_3$ twisting |
| | $\nu_{20}$ | | 385 (<0.1) | 392 (<0.1) | | C-S-N twist |
| | $\nu_{21}$ | | 321 (1.1) | 328 (1.2) | | C-S-N twist |
| | $\nu_{23}$ | | 218 (0.2) | 216 (1.5) | | $CH_3$ twist |
| | $\nu_{24}$ | | 170(14.9) | 179(11.2) | | $NH_2$ twist |
| A' | $\nu_2$ | 3380 (17.2) | 3512 (13.0) | 3508 (13.8) | 0.962 | $NH_2$ sym. stretch |
| | $\nu_4$ | | 3184 (<0.1) | 3161 (<0.1) | | $CH_3$ asym. stretch |
| | $\nu_5$ | | 3079 (0.1) | 3065 (<0.1) | | $CH_3$ sym. stretch |
| | $\nu_6$ | 1551 (15.6) | 1591 (15.2) | 1582 (13.5) | 0.975 | $NH_2$ bend |
| | $\nu_8$ | | 1460 (2.5) | 1456 (1.9) | | $CH_3$ asym. bend |
| | $\nu_9$ | 1428 (3.7) | 1363 (3.3) | 1350 (4.2) | 1.048 | $CH_3$ umbrella |
| | $\nu_{11}$ | 1172 (72.8) | 1115 (69.6) | 1135 (64.2) | 1.051 | $SO_2$ sym. stretch |
| | $\nu_{13}$ | 976 (17.8) | 994 (7.6) | 987 (8.9) | 0.982 | $CH_3$ wagging |
| | $\nu_{15}$ | 857 (43.1) | 867 (42.4) | 864 (40.8) | 0.988 | $NH_2$ wagging |
| | $\nu_{16}$ | | 704 (7.2) | 704 (4.2) | | C-S stretch |
| | $\nu_{17}$ | | 663 (81.9) | 649 (81.5) | | S-N stretch |
| | $\nu_{18}$ | | 480 (14.9) | 490 (15.8) | | $SO_2$ wagging |
| | $\nu_{19}$ | | 457 (5.1) | 468 (3.5) | | $SO_2$ bend |
| | $\nu_{22}$ | | 285 (1.8) | 290 (1.9) | | C-S-N bend |

a) Relative absorbance at band maximum in parentheses. b) Calculated using the B3LYP method.





**Table 2.** Rate coefficients ratios and experimental conditions for the relative rate experiments.

| Reference reactant [a] | Concentration ($10^{14}$ molecule cm$^{-3}$) | | | | Band[e] (cm$^{-1}$) | $k_4 / k_{\text{ref}}$ | $k_4$ ($10^{-13}$ cm$^3$ molecule$^{-1}$ s$^{-1}$) | |
|---|---|---|---|---|---|---|---|---|
| | [MSAM][b] | [Ref][c] | [O$_3$][d] | [H$_2$][c] | | | | |
| Acetone | 0.31 | 0.34 | 7.71 | 71.8 | 857 | $0.792 \pm 0.012$ | $1.43 \pm 0.10$ | $1.40 \pm 0.09$[f] |
| | | | | | 1172 | $0.771 \pm 0.005$ | $1.39 \pm 0.09$ | |
| | | | | | 1383 | $0.779 \pm 0.006$ | $1.40 \pm 0.09$ | |
| | | | | | 3380 | $0.770 \pm 0.010$ | $1.39 \pm 0.10$ | |
| Formic Acid | 0.56 | 0.55 | 5.74 | 65.2 | 857 | $0.312 \pm 0.004$ | $1.41 \pm 0.11$ | $1.38 \pm 0.09$[f] |
| | | | | | 1172 | $0.311 \pm 0.002$ | $1.40 \pm 0.10$ | |
| | | | | | 1383 | $0.302 \pm 0.002$ | $1.36 \pm 0.10$ | |
| Methanol | 0.88 | 0.25 | 4.58 | 46.6 | 1172 | $0.159 \pm 0.001$ | $1.43 \pm 0.17$ | $1.42 \pm 0.16$[f] |
| | | | | | 1383 | $0.153 \pm 0.001$ | $1.38 \pm 0.17$ | |
| | | | | | 3380 | $0.161 \pm 0.003$ | $1.45 \pm 0.19$ | |

a) Depletion of reference reactants monitored at 1221-1249, 1073-1133 and 2788-3070 cm$^{-1}$ for acetone, formic acid and methanol, respectively.
b) Concentration estimated from the absorption cross section reported in Figure 1.
c) Concentration calculated from the measured pressures,
d) Concentration derived from the absorption cross section of O$_3$
e) IR absorption bands of MSAM used for the determination of the concentration change over time,
f) Relative rate constant obtained from the linear fitting of all the data and using $k_4 = (1.8 \pm 0.1) \times 10^{-13}$, $k_5 = (4.5 \pm 0.36) \times 10^{-13}$, and $k_6 = (9.0 \pm 1.3) \times 10^{-13}$ cm$^3$ molecule$^{-1}$ s$^{-1}$ rate constants respectively (IUPAC, 2019).




**Table 3.** Lifetimes of atmospheric organo-sulphur trace gases with respect to reaction with OH

|  | $k(OH)^a$ | Lifetime[b] | Reference |
|---|---|---|---|
| $CH_3SO_2NH_2$ | $1.4 \times 10^{-13}$ | 80 days | This work |
| $CH_3SO_2CH_3$ | $< 3 \times 10^{-13}$ | $> 40$ days | (Falbe-Hansen et al., 2000) |
| $CH_3S(O)CH_3$ | $5.9 \times 10^{-11}$ | 5 hours | (Falbe-Hansen et al., 2000) |
| $CH_3SO_2H$ | $9.0 \times 10^{-11}$ | 2.8 hours | (Burkholder et al., 2015) |
| $CH_3SCH_3$ | $2.2 \times 10^{-12}$ | 1.6 days | (Atkinson et al., 2004) |

[a] Units of $cm^3$ molecule$^{-1}$ s$^{-1}$. [b] Assumes a diel average OH concentration of $1 \times 10^6$ molecule $cm^{-3}$.




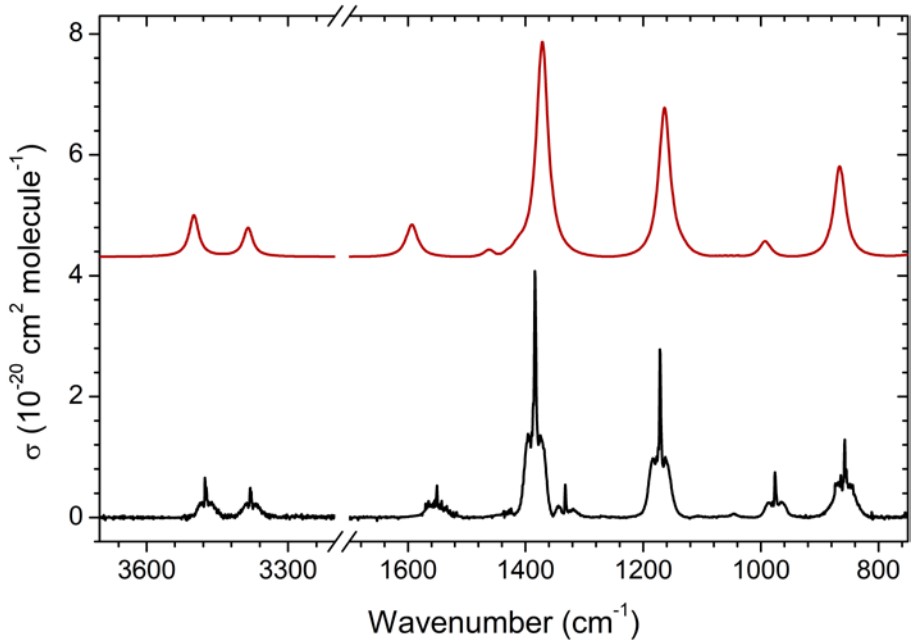

**Figure 1.** Comparison of the experimental infra-red absorption spectrum (black line) of MSAM (~1 × 10$^{14}$ molecule cm$^{-3}$) and the calculated spectrum at B3LYP/aug-cc-pVTZ-pp levels of theory (red line). The cross-sections (σ) are in base e.





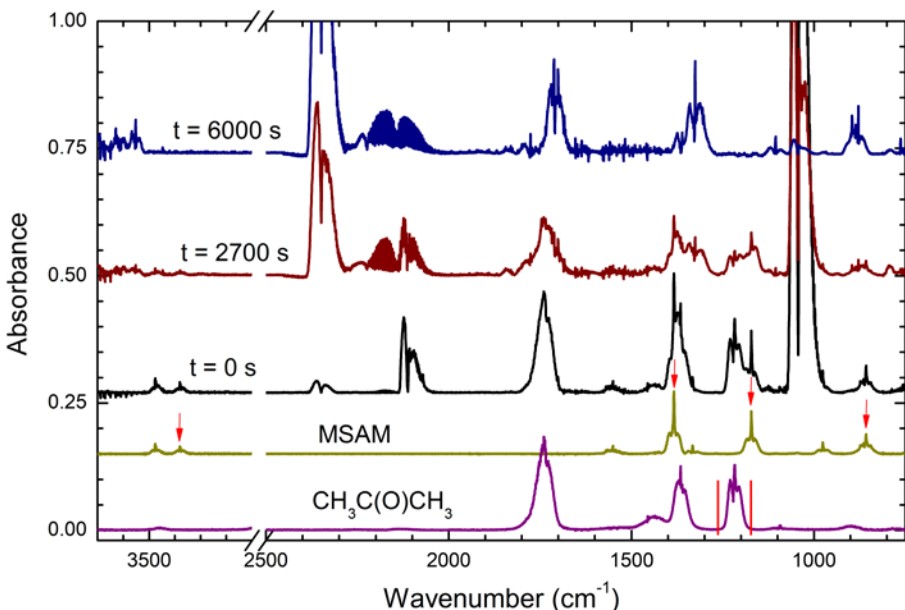

**Figure 2.** Raw data from a relative rate experiments (using acetone as reference reactant) showing the gradual depletion of reactants and formation of products. The lower two spectra are reference spectra, indicating (red arrows and red min-max lines) which absorption features were used for the relative rate analysis. The strong absorption close to 2100 cm$^{-1}$ at t = 0 s is due to $O_3$. $CO_2$ and CO absorption bands are centered at $\approx$ 2350 and 2140 cm$^{-1}$, respectively.





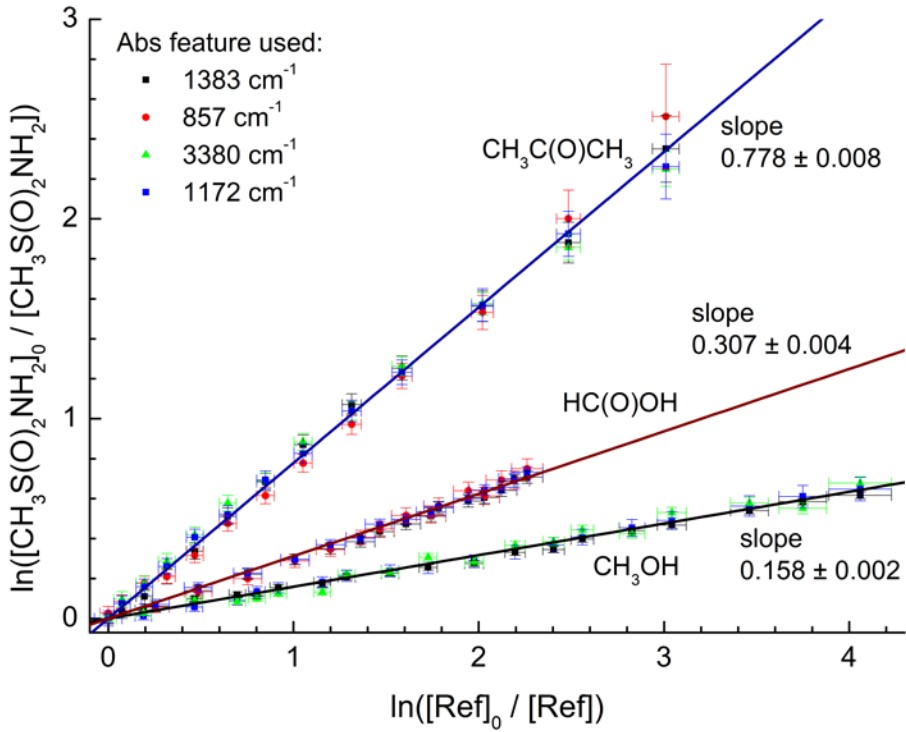

**Figure 3.** Relative depletion factors for MSAM and the reference compounds $CH_3C(O)CH_3$, $HC(O)OH$, and $CH_3OH$ obtained at room temperature and a total pressure of 700 Torr of air. The different absorption bands of MSAM used in the analysis are indicated. The slopes are equal to the ratio of rate coefficients $k_4 / k_{ref}$ as defined in Eqn. 1).




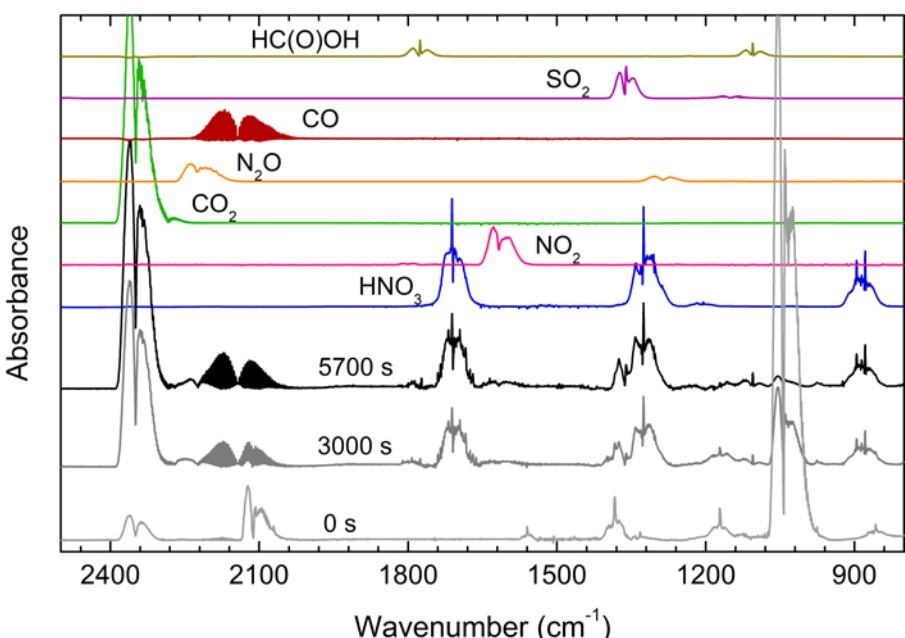

**Figure 4.** Infrared spectrum acquired prior to (A) and after (B and C) photolysis of a mixture of MSAM, $O_3$, $H_2$ and air. During irradiation, OH concentrations were $\approx 3 \times 10^9$ molecule $cm^{-3}$. $H_2O$-absorption features have been subtracted from the spectra. Reference spectrum of $HNO_3$, $NO_2$, $CO_2$, $N_2O$, $CO$, $SO_2$, and $HC(O)OH$ recorded under the same experimental conditions (700 Torr and 298 K) are also shown.





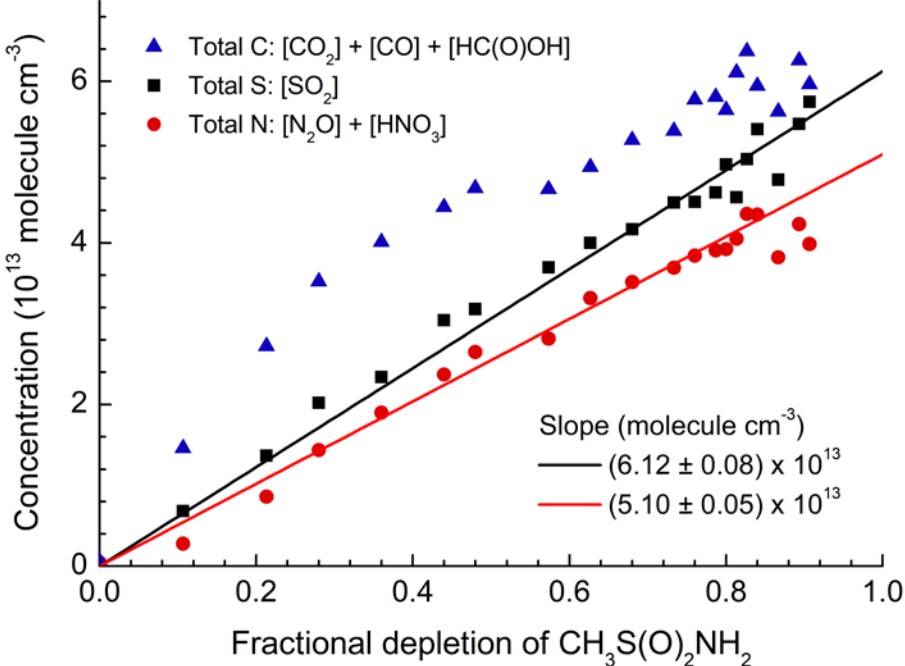

**Figure 5.** Summed concentration of carbon containing products (CO₂, CO and HC(O)OH), sulphur containing products (SO₂) and nitrogen containing products (N₂O and HNO₃) plotted versus the relative depletion of MSAM. The slope gives the initial concentration of MSAM assuming stoichiometric conversion to the products listed.

.






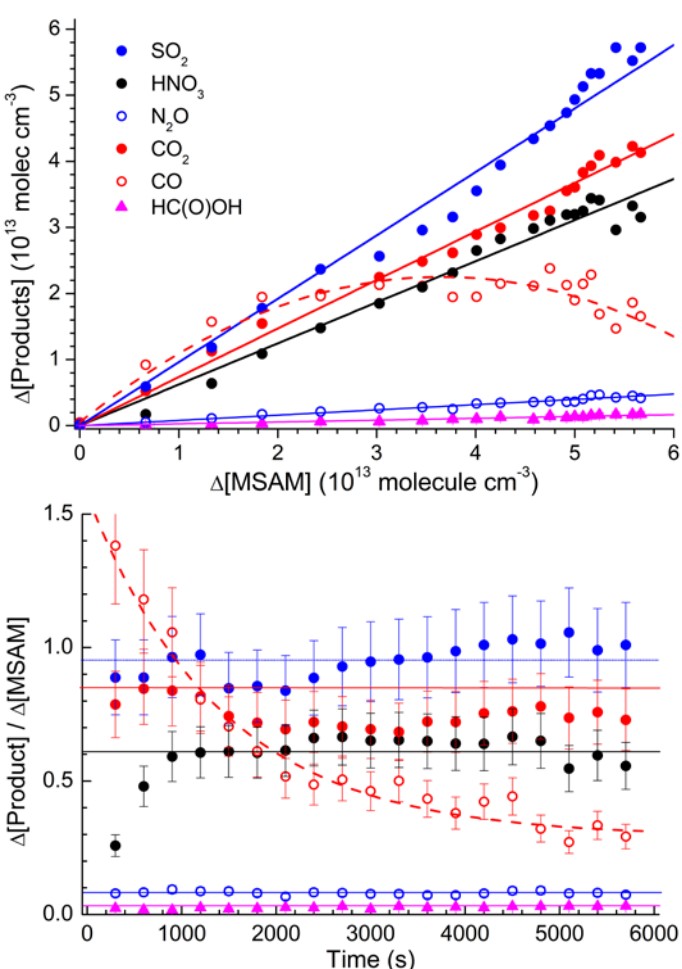

**Figure 6.** Upper: Formation of products versus depletion of MSAM. The slopes of the linear fits are the yields of $SO_2$, $HNO_3$, $CO_2$, $N_2O$ and $HC(O)OH$ from this particular experiment. The polynomial fit to the CO data (red dash line) is added to guide the eye. Lower: Time dependence of the product yields from the same experiment. The solid, horizontal lines represent the average yield. Error bars are total estimated uncertainty ($2\ \sigma$) including uncertainty in cross-sections of products and MSAM.








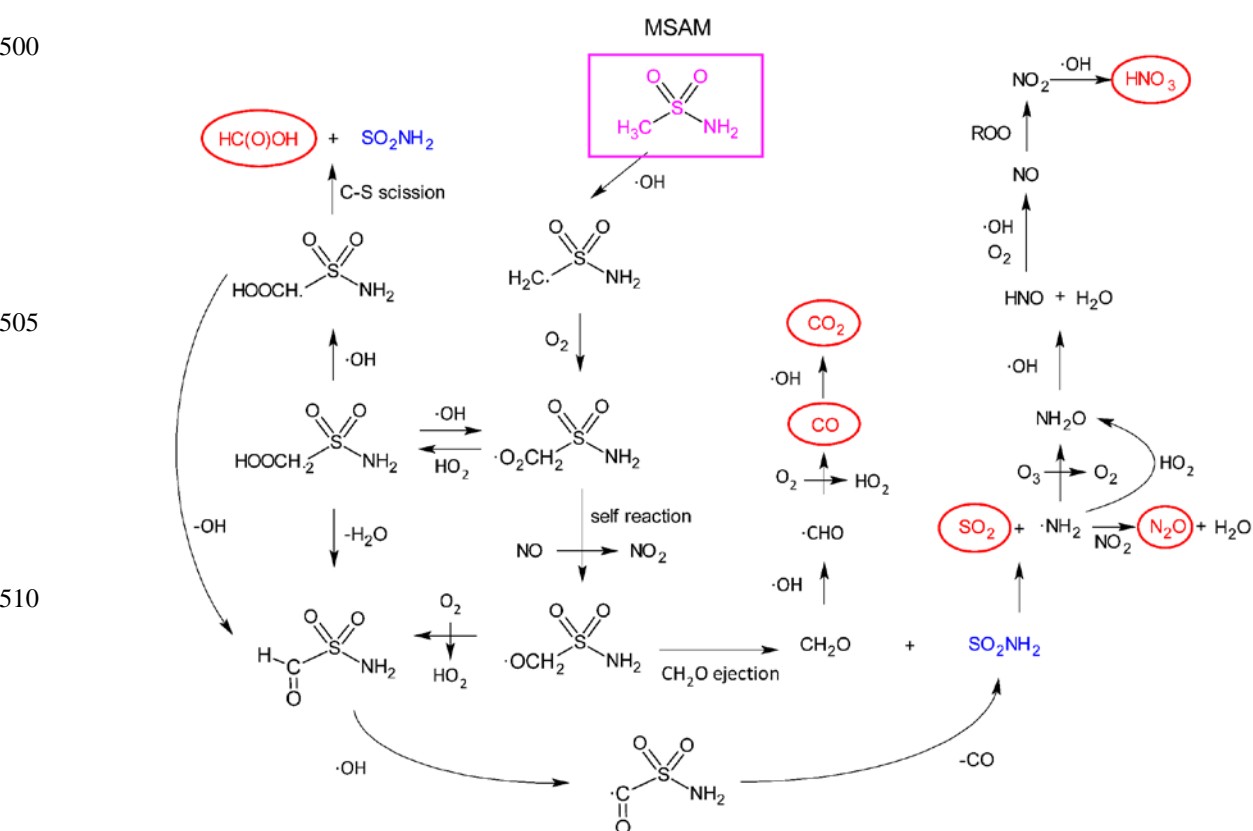

**Figure 7.** Mechanism for the formation of carbon containing (HC(O)OH, CO, CO₂), nitrogen containing (HNO₃ and N₂O) and sulphur containing (SO₂) end products (in red circles) observed in the OH-initiated photo-oxidation of MSAM.



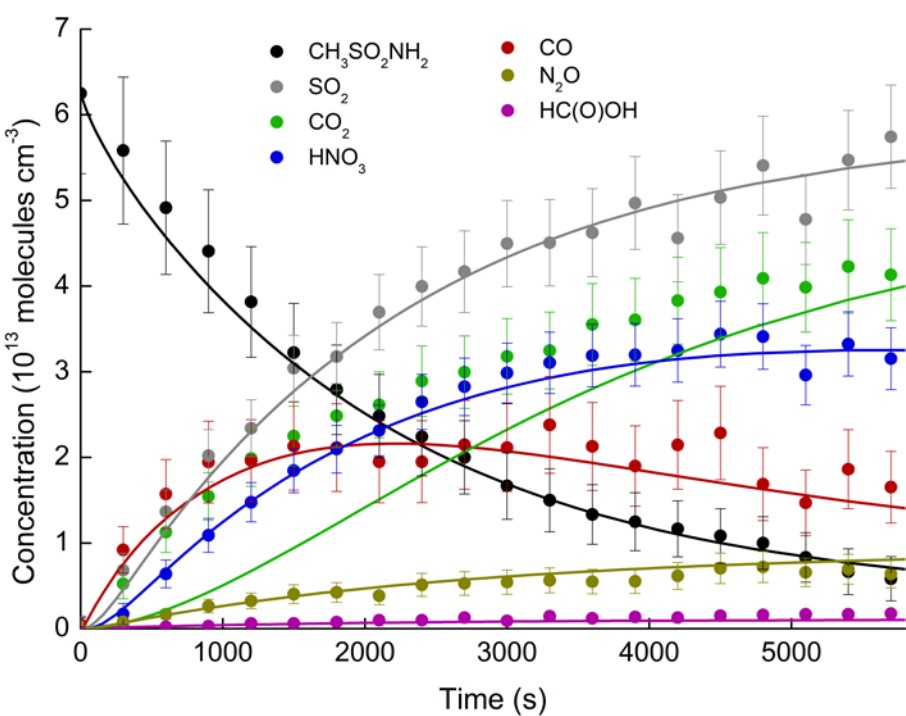

**Figure 8.** Concentration–time profiles of MSAM and reaction products formed in an experiment at 298 K and 700 Torr air. The error bars represent uncertainty in infra-red cross sections. The uncertainty associated with the MSAM concentration is estimated to be $\approx 25\%$. Solid lines are simulated profiles.