# Peer review of "Kinetic and mechanistic study of the reaction between methane sulphonamide ( $\text{CH}_3\text{S(O)}_2\text{NH}_2$ ) and OH"

_Atmospheric Chemistry and Physics, 2019_

## Referee Comment (RC1) · Anonymous Referee #1 · 13 Dec 2019

This paper reports the results of a thorough study of the atmospheric chemistry of methane sulphonamide. I recommend publication as is.

---

## Referee Comment (RC2) · Anonymous Referee #2 · 18 Dec 2019

This paper describes original laboratory studies of the OH rate coefficient and atmospheric oxidation pathways for methane sulphonamide (MSAM), a molecule recently detected for the first time in ambient air by a subset of the co-authors. In general, this is an excellent study, and the subject of the work is clearly suitable for publication in ACP. Assumptions and limitations are clearly and logically presented and discussed. It is comforting to see detailed calibration curves presented in the supplement, kinetic simulations of the proposed mechanism carried out, etc. The observation of an anomalous rate of oxidation of SO2 by OH is intriguing, and I look forward to hearing more about this issue in future work. Some questions and comments are given below, the first one of more significance than the remainder. It is my opinion that the paper will be

publishable in ACP after these comments are addressed.

Main question: I did not get a good sense of how reproducible the MSAM IR cross sections or the product yields really are. How many experiments were actually carried out, and how were final values and uncertainties obtained? (At times, the section 3.4 reads as though one experiment only was done, which would not seem sufficient to me).

Minor comments and questions: I think most of the experiments (kinetics and product studies) were done in air – if so, it would help to state this early on (around line 45?).

Line 59 – space needed between 'and' and 'SO2'.

Line 75 – Initial H2 concentrations shown in Table 1 are lower than the 5e16 molecule cmˆ-3 value given here.

Line 135 - The existence of an MSAM dimer hits the reader rather abruptly. I suggest discussing the existence of the dimer in more detail in the experimental setup section – how the dimer was identified, its elimination with the cold-trap, etc.

Line 164 – 'different' spelled incorrectly.

Line 170 – product yield studies: concentrations of MSAM employed are quite a bit lower than the vapour pressure upper limit given earlier. Could higher [MSAM] have been used to limit the extent of 2ry chemistry, and get a better handle on the initial products formed?

Line 185 – I think this should be HNO3 + 2*N2O?

Line 209 – I would argue that, given the complex time profile for CO, giving a yield for this species is not meaningful, and potentially misleading. It might be also worth reiterating somehow that HNO3 and CO2 are secondary products, and so the yields really only apply after significant loss of MSAM has occurred.

Section 3.4.1 – I am assuming that NO is not seen in any of the IR spectra? Is this

consistent with the mechanism, and the NO detection limit? Also, can more be said about whether the NO2 temporal profile observed matches the mechanistic predictions? Can the authors be more quantitative about the CH2O steady-state seen in the model, compared with its IR detection limit?

Line 376 – s should be sˆ-1.

Table 1 – Footnote b should only refer to the theoretical headers, not the experimental one?

Figure 6, lower trace: All the CO2 measurements fall below the average line. Please clarify.

Figure 7: Is there any chance that the radical down at the bottom of the scheme, O=CS(O)(O)NH2, adds O2 rather than decomposing? This would probably lead to a CO2 rather than a CO?

Also, it may end up not changing anything, but there could also be a 'molecular channel' to the peroxy radical self-reaction?

Table S2: It doesn't matter to the final result, but a decomposition of 10ˆ25 s-1 for the alkoxy species doesn't seem possible to me.

Maybe not critical, but the HO2 self-reaction is not included in the mechanism.

NHOH seems likely to be a fairly reactive radical – are there other possible losses for this species that are not included in the mechanism? Reaction with O2 to make HNO and HO2??

---

## Author Comment (AC1) · 24 Jan 2020

We thank the referee for this positive assessment of our manuscript.

———————————————

---

## Author Response (AR1)

The following contains the comments of the referee (black) and our replies (blue) indicating changes made to the revised document (red).

**Referee #2**

This paper describes original laboratory studies of the OH rate coefficient and atmospheric oxidation pathways for methane sulphonamide (MSAM), a molecule recently detected for the first time in ambient air by a subset of the co-authors. In general, this is an excellent study, and the subject of the work is clearly suitable for publication in ACP. Assumptions and limitations are clearly and logically presented and discussed. It is comforting to see detailed calibration curves presented in the supplement, kinetic simulations of the proposed mechanism carried out, etc. The observation of an anomalous rate of oxidation of SO2 by OH is intriguing, and I look forward to hearing more about this issue in future work. Some questions and comments are given below, the first one of more significance than the remainder. It is my opinion that the paper will be publishable in ACP after these comments are addressed.

We thank the referee for this positive assessment of our manuscript.

Main question: I did not get a good sense of how reproducible the MSAM IR cross sections or the product yields really are. How many experiments were actually carried out, and how were final values and uncertainties obtained? (At times, the section 3.4 reads as though one experiment only was done, which would not seem sufficient to me).

A total of 4-experiments were evaluated to obtain the cross-sections of MSAM. We now indicate this in the manuscript and modify a figure in the supplementary information (Fig. S3) to include data from all 4 experiments. The text now reads:

Figure S3 of the supplementary information illustrates the strict proportionality between the relative change of the SO2 concentration and the MSAM absorption feature at 1384 cm-1 in 4 different experiments. From these 4 experiments we derive an absorption cross-sections for MSAM at this wavenumber of  $(4.06 \pm 0.17) \times 10^{-19}$  cm2 molecule-1. This value was used to scale the spectrum of MSAM (Fig. 1) and was used to calculate initial concentrations in all other experiments.

We also noticed that the Figure 1 had the wrong exponent (20 instead of 19) on the y-axis. This has been rectified.

Minor comments and questions: I think most of the experiments (kinetics and product studies) were done in air – if so, it would help to state this early on (around line 45?). We now state:

The quartz reactor was at room temperature  $(296 \pm 2 \text{ K})$  and for most experiments at 700 Torr total pressure (1 Torr = 1.333 hPa) using synthetic air bath gas.

Line 59 – space needed between 'and' and 'SO2'. Correction made

Line 75 – Initial H2 concentrations shown in Table 1 are lower than the 5e16 molecule cm-3 value given here.

This was a typo. We now write: In a typical experiment, the starting concentrations of O3 and H2 were  $\approx 5 \times 10^{14}$  and  $\approx 5-7 \times 10^{15}$  molecule cm-3.

Line 135 - The existence of an MSAM dimer hits the reader rather abruptly. I suggest discussing the existence of the dimer in more detail in the experimental setup section – how the dimer was identified, its elimination with the cold-trap, etc.

The text in the experimental section has been modified to mention the dimer:

Owing to its low vapour pressure, MSAM was eluted into the reaction chamber by flowing synthetic air (450 cm3 STP min-1, sccm) through a finger containing crystalline MSAM warmed to 333 K, and subsequently through a cold trap at 298 K (to prevent condensation downstream). This way we could ensure that the saturation vapour pressure of MSAM at 298 K was achieved. In initial experiments without the trap we observed extra absorption features, which could be assigned to a dimer of MSAM (see below).

Line 164 – 'different' spelled incorrectly. Correction made.

Line 170 – product yield studies: concentrations of MSAM employed are quite a bit lower than the vapour pressure upper limit given earlier. Could higher [MSAM] have been used to limit the extent of 2ry chemistry, and get a better handle on the initial products formed?

We do not give a vapour pressure of MSAM (only an upper limit) as we were not able to directly measure the pressure of the vapour above MSAM crystals at 298 K. We always operated under conditions of maximum MSAM eluting from the trap at 298 K. WE now write:

Crystalline MSAM melts at 363 K, has a boiling point of approximately 453 K and an unknown vapour pressure (< 0.02 Torr) at room temperature.

Line 185 - I think this should be HNO3 + 2\*N2O?

This is true. We now write:

Figure 5 plots the concentration of  $SO_2$  (the only sulphur containing product observed), the sum of  $HNO_3 + 2 N_2O$  (the total reactive nitrogen observed as product)

Line 209 – I would argue that, given the complex time profile for CO, giving a yield for this species is not meaningful, and potentially misleading. It might be also worth reiterating somehow that HNO3 and CO2 are secondary products, and so the yields really only apply after significant loss of MSAM has occurred.

We no longer list the yield of CO and mention that the yields are derived for 80% MSAM depletion.

In contrast, the CO yield is initially larger than unity (indicative of extra sources from the chamber walls) and then decreases with time as it is removed (via reaction with OH) to form CO2.

The molar yields (after 6000 seconds of photolysis when MSAM has depleted to  $\sim$ 20% of its original concentration) of the products obtained at 298 K and 700 Torr of synthetic air are:

 $\Phi(SO_2) = 0.96 \pm 0.15, \ \Phi(HNO_3) = 0.62 \pm 0.09, \ \Phi(N_2O) = 0.09 \pm 0.02, \ \Phi(CO_2) = 0.73 \pm 0.11, \ \Phi(HC(O)OH) = 0.03 \pm 0.01.$

Section 3.4.1 – I am assuming that NO is not seen in any of the IR spectra? Is this consistent with the mechanism, and the NO detection limit? Also, can more be said about whether the NO2 temporal profile observed matches the mechanistic predictions? Can the authors be more quantitative about the CH2O steady-state seen in the model, compared with its IR detection limit?

We now explain why we did not detect NO, NO2 or HCHO:

The simulations indicate that the maximum concentrations of NO (7 × 109 molecule cm-3) and NO2 (~1012 molecule cm-3) are below the detection limit of the instrument, and were therefore not observed. The strongest absorption features of HCHO (1700-1800 cm-1) overlap with those of H2O and HNO3 so that the predicted concentrations of HCHO (< 1012 molecule cm-3) are also below the detection limit.

Line 376 - s should be  $s^{-1}$ . Correction made.

Table 1 – Footnote b should only refer to the theoretical headers, not the experimental one? Correction made.

Figure 6, lower trace: All the CO2 measurements fall below the average line. Please clarify. This was a mistake; the yield-line for  $CO_2$  had not been re-drawn after scaling the plot. This has been corrected as shown below

Figure 7: Is there any chance that the radical down at the bottom of the scheme, O=CS(O)(O)NH2, adds O2 rather than decomposing? This would probably lead to a CO2 rather than a CO?

This may also happen, though it probably also ends up as  $SO_2NH_2$  and does not change anything as far as the sulphur and nitrogen containing products are concerned. Nonetheless, we have added this reaction to the scheme in Figure 7 and added some text.

 $C(O)SO_2NH_2$  may either decompose to  $SO_2NH_2$  and CO (R17) or react with  $O_2$  to form a  $\alpha$ -carbonyl peroxy radical (R18).

 $\begin{array}{ccc} C(O)SO_2NH_2 & \rightarrow & SO_2NH_2 + CO & (R17) \\ C(O)SO_2NH_2 + O_2 + M & \rightarrow & O_2C(O)SO_2NH_2 + M & (R18) \end{array}$

The fate of  $O_2C(O)SO_2NH_2$  is likely to be dominated by reaction with HO2 which, by analogy to  $CH_3C(O)O_2$  (another  $\alpha$ -carbonyl peroxy radical) is expected to lead to the reformation of OH (Dillon and Crowley, 2008; Groß et al., 2014).  $O_2C(O)SO_2NH_2 + HO_2$  $OH + O_2 + CO_2 + SO_2NH_2$ (R19)  $\rightarrow$

In both scenarios, SO2NH2 is the sulphur containing product, whereas formation of the peroxy radical will result in prompt CO2 formation and OH-recycling.

We also included this reaction in the numerical simulations. It results in the production of prompt CO2 (which better matches the observations) but CO is no longer formed (contrary to the observations). We now write:

The simulation also captures the CO profile well, but fails to predict the early formation of CO2. The match between simulation and experiment could be improved to some extent for CO2 by amending the fate of the C(O)SO2NH2 radical as described above (R17-R18) so that CO2 rather than CO is formed. The results (Fig S5 of the supplementary information) indicate that the improved simulation of CO2 is accompanied by complete loss of agreement with CO (which is no longer formed in measurable amounts) and poorer agreement with e.g. SO2 and HNO3. However, given that CO2 is generated from the cell walls during irradiation and cannot be used quantitatively (Section 3.5), the fate of the  $C(O)SO_2NH_2$  radical remains obscure. We emphasise that the reproduction of the profiles of the observed end products does not constitute quantitative understanding of the fate of several radical and non-radical intermediates formed.

We have added an extra plot the SI to exemplify this.

Fig S5

Also, it may end up not changing anything, but there could also be a 'molecular channel' to the peroxy radical self-reaction?

We cannot rule out that NH2SO2CH2O2 can self-react to make an alcohol and aldehyde (as well as the alkoxy channel listed). Simulations showed that this reaction had little effect, presumably because the high levels of HO2 mean that most RO2 react with HO2 and not with themselves.

Table S2: It doesn't matter to the final result, but a decomposition of 1025 s-1 for the alkoxy species doesn't seem possible to me.

As indicated in the footnotes this is a theoretical value and not expected to be accurate. For simplicity we now write  $> 1 \times 9$  s-1 and note that any value above a few s-1 would have the same effect.

fCalculated with G4MP2 level of theory. As this is the only reaction of OCH2SO2NH2 that we consider, any lifetime shorter than a few seconds would give the same simulation result.

Maybe not critical, but the HO2 self-reaction is not included in the mechanism.

The self-reaction of  $HO_2$  was included in the simulations, but we did not list it in Table S2 as its inclusion had no significant impact on the results.

To make the reaction scheme more complete, we now list the  $HO_2$  self-reaction (and several other reactions that we had previously neglected to include) in Table S2.

NHOH seems likely to be a fairly reactive radical – are there other possible losses for this species that are not included in the mechanism? Reaction with O2 to make HNO and HO2?? The Referee is correct. The reaction of NHOH with O2 was considered (along with OH) in the reaction mechanism and, with a rate constant =  $9.6 \times 10^{-12}$  cm3 molecule-1 s-1, it is the dominant loss process.

Table S2 has been modified accordingly.

**Kinetic and mechanistic study of the reaction between methane sulphonamide (CH3S(O)2NH2) and OH**

Matias Berasategui1, Damien Amedro1, Achim Edtbauer1, Jonathan Williams1, Jos Lelieveld1 and John N. Crowley1

[revised manuscript text omitted]

Further reactions that cycle OH and HO2 (e.g. OH +  $H_2$ , H + O3, HO2 + O3) are listed in Table S1 of the supplementary information.

In a typical experiment, the starting concentrations of  $O_3$  and  $H_2$  were  $\approx 5 \times 10^{14}$  and  $\approx 5.7 \times 10^{15}$  molecule cm-3. As described previously (Bunkan et al., 2018), this scheme generates not only OH radicals but (via e.g. (R3) also HO2. HO2 is not expected to react with MSAM but will influence the course of secondary reactions in this system (e.g. by reacting with organic peroxy radicals) and thus the end-product distribution, as described in detail in section 3.5. Simulations of the radical concentrations when generating OH in this manner indicate that the HO2 / OH ratio is approximately 30, with individual

so concentrations when generating OH in this manner indicate that the HO2 / OH ratio is approximately 30, with individual concentrations of  $\approx 1 \times 10^{11}$  molecule cm-3 HO2 and  $3 \times 10^{9}$  molecule cm-3 OH.

As OH source, the photolysis of  $O_3$  in the presence of  $H_2$  has the advantage over other photochemical sources (e.g. photolysis of  $H_2O_2$ , HONO or CH3ONO) that neither  $H_2$  nor  $O_3$  have strong absorption features in the infra-red, resulting in a less cluttered spectrum which simplifies retrieval of concentration-time profiles of reactants and products.

**85 2.3 Chemicals**

A commercially available sample of methane sulphonamide (Alfa Aesar, > 98%) was used. O3 was generated by flowing synthetic air (Westfalen) through a stainless steel tube that housed a low-pressure Hg-lamp (PenRay) emitting at 184.95 nm. Synthetic air (Westfalen, 99.999%), H2 (Westfalen, 99.999%), CO2 (Westfalen 99.995%), CO (Westfalen, 99.997%), SO2 (Air Liquide, 1 ppmv in air) and HC(O)OH (Sigma Aldrich) were obtained commercially. Anhydrous nitric acid was

90 prepared by mixing KNO3 (Sigma Aldrich, 99%) and H2SO4 (Roth, 98%), and condensing HNO3 vapour into a liquid nitrogen trap.

**2.4 Relative rate constant determination**

95

The rate constant ( $k_4$ ) of the reaction between OH and CH3SO2NH2 (Reaction R4) was measured using the relative rate method using (in different experiments) formic acid (HC(O)OH), acetone (CH3C(O)CH3) and methanol (CH3OH) as reference compounds.

| $CH_3S(O)_2NH_2 + OH$ | $\rightarrow$ | Products | (R4) |
|-----------------------|---------------|----------|------|
| $CH_3C(O)CH_3 + OH$   | $\rightarrow$ | Products | (R5) |
| HC(O)OH + OH          | $\rightarrow$ | Products | (R6) |
| $CH_{3}OH + OH$       | $\rightarrow$ | Products | (R7) |

100 Relative rate constants were derived by monitoring the depletion of one or more IR-features of MSAM relative to those of the reference compounds. The following expression links the depletion factors (e.g. ln([MSAM]0/[MSAM]t) to the relative rate coefficient:

$$\ln\left(\frac{[\text{MSAM}]_0}{[\text{MSAM}]_t}\right) = \frac{k_4}{k_{\text{ref}}} \ln\left(\frac{[\text{ref}]_0}{[\text{
[revised manuscript text omitted]
 CO2 and CO, nitric acid (HNO3) and sulphur dioxide (SO2) are easily identified, with weak features from N2O, NO2 and formic acid (HC(O)OH) also apparent. The absorption of each product was converted to a concentration using calibration curves that were obtained at the same pressure and temperature (see Fig. S2 of the supplementary information).
- Figure 5 plots the concentration of SO2 (the only sulphur containing product observed), the sum of  $HNO_3 + 2 N_2O$  (the total reactive nitrogen observed as product) and the sum of  $CO_2 + CO + HC(O)OH$  (total carbon containing products observed)

versus the fractional depletion of MSAM. The concentrations after 6000 seconds (when ~90% of the MSAM had reacted) were:  $[SO_2] = 5.74 \times 10^{13}$ ,  $[HNO_3] = 3.15 \times 10^{13}$ ,  $[N_2O] = 4.17 \times 10^{12}$ ,  $[CO_2] = 4.13 \times 10^{13}$ ,  $[CO] = 1.65 \times 10^{13}$  and  $[HC(O)OH] = 1.77 \times 10^{12}$  molecules cm-3. MSAM contains one atom each of sulphur, nitrogen and carbon. If SO2, reactive

- 190 nitrogen and carbon are conserved, we can derive initial concentrations of MSAM (from the slope) of  $6.12 \pm 0.08 \times 10^{12}$ molecule cm-3 (based on the sulphur balance),  $5.10 \pm 0.05 \times 10^{12}$  molecule cm-3 (based on the nitrogen balance) and  $7.4 \pm 0.2$  $\times 10^{12}$  molecule cm-3 (based on the carbon balance at the maximum fractional depletion of MSAM). As already mentioned, total carbon is very likely to be overestimated due to its formation and desorption at/from the walls of the chamber. As the main nitrogen product is HNO3, which has a large affinity for surfaces and which is likely to be lost to the walls, we also
- 195 expect that use of reactive nitrogen will result in an underestimation of the initial MSAM concentration. For these reasons we consider that the best method to estimate the initial concentration of MSAM is via the formation of SO2. Figure S3 of the supplementary information illustrates the strict proportionality between the relative change of the SO2 concentration and the MSAM absorption feature at 1384 cm-1 in 4 different experiments. From these 4 experiments we derive an absorption crosssections for MSAM at this wavenumber of  $(4.06 \pm 0.17) \times 10^{-19}$  cm2 molecule-1. This value was used to scale the spectrum of
- 200 MSAM (Fig. 1) and was used to calculate initial concentrations in all other experiments. Figure 6a presents a plot of  $\Delta$ [product] versus - $\Delta$ [MSAM] from one experiment. Apart from CO, we observe a roughly linear relationship for all products. Time dependent yields of each product are displayed in Fig. 6b. Whereas the yields of SO2, CO2, N2O and HC(O)OH are, within experimental scatter roughly constant, that of HNO3 (black line) reaches a constant value only after 800 seconds, indicating that it is not formed directly, but in a secondary reaction. In contrast, the CO yield is initially slightly
- 205 larger than unity (indicative of extra sources from the chamber walls) and then decreases with time as it is removed (via reaction with OH) to form CO2. The molar yields (after 6000 seconds of photolysis when MSAM has depleted to ~20% of its original concentration) of the products obtained at 298 K and 700 Torr of synthetic air are:  $\Phi(SO_2) = 0.96 \pm 0.15$ ,  $\Phi(HNO_3) = 0.62 \pm 0.09$ ,  $\Phi(N_2O) = 0.09 \pm 0.02$ ,  $\Phi(CO_2) = 0.73 \pm 0.11$ ,  $\Phi(HC(O)OH) = 0.03 \pm 0.01$ . The slight deviation of  $\Phi(SO_2)$  from unity stems from the fact that the quoted yield is at a fixed time, whereas the initial MSAM concentration was derived using all the SO2 data in this experiment as described above. As N2O contains two N-atoms, the nitrogen balance is thus 0.80
- $\pm$  0.13. It is likely that some HNO3 is lost to reactor surfaces, explaining the deviation from unity. Note that if we had used the nitrogen balance to derive the MSAM IR-cross-sections, the SO2 yield would have exceeded unity.

**3.4 Reaction mechanism**

The time dependent formation of HNO3, SO2, N2O and CO provide important clues to the reaction mechanism. Addition to 215 the S-atom is not possible so that the initial step will be abstraction of hydrogen by the OH radical, either from the -CH3 group (Reaction R8a) or from the -NH2 group (Reaction R8b):

$$CH_{3}SO_{2}NH_{2} + OH \longrightarrow CH_{2}SO_{2}NH_{2} + H_{2}O$$

$$(R8a)$$

$$CH_{3}SO_{2}NH_{2} + OH \longrightarrow CH_{3}SO_{2}NH_{2} + H_{2}O$$

$$(R8b)$$

$$H_3SO_2NH_2 + OH \longrightarrow CH_3SO_2NH + H_2O$$
(R8b)

Based on results of previous studies of the reactions of OH with trace-gases containing both CH3 and -NH2 entities (e.g.

220 CH3NH2 or CH3C(O)NH2) we expect abstraction at the –CH3 group (Reaction R8a) to dominate (Onel et al., 2014; Borduas et al., 2015; Butkovskaya and Setser, 2016). H-abstraction at the methyl-group is also consistent with a rate coefficient for R4 that is very similar to that for OH + acetone.

**3.4.1** Abstraction from the -CH3 group**

In section 3.4.1 we focus on the fate of the peroxy radical, OOCH2SO2NH2, formed by reaction of initially formed 225 CH2SO2NH2 with O2 (R9). The most important reactions of organic peroxy radicals are self-reactions (R10) or reactions with NO (R11), NO2 (R12), or HO2 (R13).

| $CH_2SO_2NH_2+O_2\\$                                | $\rightarrow$ | OOCH 2 SO 2 NH 2 | (R9)  |
|-----------------------------------------------------|---------------|---------------------------------------------------|-------|
| 2 OOCH 2 SO 2 NH 2 | $\rightarrow$ | $OCH_2SO_2NH_2 + O_2$                             | (R10) |
| $OOCH_2SO_2NH_2 + NO$                               | $\rightarrow$ | $OCH_2SO_2NH_2 + NO_2$                            | (R11) |

 $\begin{array}{rcl} 230 & OOCH_2SO_2NH_2 + NO_2 & \rightarrow & O_2NOOCH_2SO_2NH_2 \\ & OOCH_2SO_2NH_2 + HO_2 & \rightarrow & HOOCH_2SO_2NH_2 + O_2 \end{array} \tag{R12}$

Peroxy nitrates such as the one formed in Reaction (R12) are thermally unstable with respect to dissociation back to reactants at room temperature and given the very low concentrations of  $NO_2$  in our system, Reaction (R12) will not play a significant role in this study.

The oxy-radical, OCH2SO2NH2 formed in Reactions (R10) and (R11) will react with O2 to produce an aldehyde (Reaction R14). Alternatively, it could undergo C-S bond cleavage (Reaction R15) to form formaldehyde (CH2O) and the SO2NH2 radical:

 $OCH_2SO_2NH_2 + O_2 \longrightarrow HC(O)SO_2NH_2 + HO_2$ (R14)

 $OCH_2SO_2NH_2 \longrightarrow CH_2O + SO_2NH_2$ (R15)

240 The fate of HC(O)SO2NH2 will be reaction with OH to form C(O)SO2NH2 (R16) which may dissociate to form CO + SO2NH2 (R17). The rate coefficient for reaction (R16) is expected to be  $\approx 10^{-11}$  cm3 molecule-1 s-1 as for many similar reactions of OH with aldehydes (e.g. CH3CHO).

$$HC(O)SO_2NH_2 + OH \rightarrow C(O)SO_2NH_2 + H_2O$$
(R16)

 $C(O)SO_2NH_2$  may either decompose to  $SO_2NH_2$  and CO (R17) or react with  $O_2$  to form a  $\alpha$ -carbonyl peroxy radical (R18).

245  $C(O)SO_2NH_2 \rightarrow SO_2NH_2 + CO$  (R17)  $C(O)SO_2NH_2 + O_2 + M \rightarrow O_2C(O)SO_2NH_2 + M$  (R18)

The fate of  $O_2C(O)SO_2NH_2$  is likely to be dominated by reaction with HO2 which, by analogy to  $CH_3C(O)O_2$  (another  $\alpha$ carbonyl peroxy radical) is expected to lead to the reformation of OH (Dillon and Crowley, 2008; Groß et al., 2014).  $O_2C(O)SO_2NH_2 + HO_2 \rightarrow OH + O_2 + CO_2 + SO_2NH_2$  (R19) 250 In both scenarios, SO2NH2 is the sulphur containing product, whereas formation of the peroxy radical in R18 will result in early CO2 formation and OH-recycling.

Formaldehyde formed in R15 will react with OH to form CO and subsequently CO2:

265

|     | $CH_2O + OH$               | $\rightarrow$ | $HCO + H_2O$      | (R20) |
|-----|----------------------------|---------------|-------------------|-------|
|     | $HCO + O_2$                | $\rightarrow$ | $HO_2 + CO$       | (R21) |
| 255 | $CO + OH (+O_2)$           | $\rightarrow$ | $CO_2 + HO_2$     | (R22) |
|     | But it may also react with | $HO_2$ to f   | form formic acid: |       |
|     | $CH_2O+HO_2+M\\$           | $\rightarrow$ | $HOCH_2OO + M$    | (R23) |
|     |                            |               |                   |       |

| $2 \text{ HOCH}_2\text{OO} \rightarrow$ | $HOCH_2O + O_2$ | (R24) |
|-----------------------------------------|-----------------|-------|
|-----------------------------------------|-----------------|-------|

 $HOCH_2O + O_2 \rightarrow HC(O)OH + HO_2$  (R25)

260 The above reactions explain, at least qualitatively, the observed formation of CO, CO2 and HC(O)OH. Note that the room temperature rate coefficient for reaction of OH with HCHO is large ( $8.5 \times 10^{-12}$  cm3 molecule-1 s-1, Atkinson et al. (2006)) compared to that for reaction with CO ( $2.2 \times 10^{-13}$  cm3 molecule-1 s-1 Atkinson et al. (2006)), which explains why CO was observed as an intermediate product at high concentrations whereas HCHO was not.

The likely fate of the  $SO_2NH_2$  radical formed in Reaction (R15) is either reaction with  $O_2$  to generate  $SO_2NH$  or dissociation by S-N bond-scission to produce  $SO_2$  and the  $NH_2$  radical.

$$SO_2NH_2 + O_2 \longrightarrow SO_2NH + HO_2$$
(R26)
$$SO_2NH_2 \longrightarrow SO_2 + NH_2$$
(R27)

We did not observe features in the IR- spectrum that that could be assigned to  $SO_2NH$  based on the spectrum reported by Deng et al. (2016) and propose that reaction (R27) is the source of  $SO_2$  as a major reaction product. By analogy with the

270 thermal decomposition of the similar CH3SO2 radical, which dissociates to CH3 and SO2 on a millisecond time scale (Ray et al., 1996), we expect SO2NH2 to decompose stoichiometrically to SO2 and NH2 on the time scale of our experiments. The NH2 radical, is known to react with O3, HO2 and NO2 (IUPAC, 2019):

$$NH_2 + O_3 \qquad \rightarrow \qquad NH_2O + O_2 \tag{R28}$$

$$NH_2 + HO_2 \rightarrow NH_2O + OH$$
 (R29a)

$$\quad NH_2 + HO_2 \qquad \rightarrow \qquad HNO + H_2O \tag{R29b}$$

$$NH_2 + NO_2 \rightarrow N_2O + H_2O$$
 (R30)

NH2O rearranges within ~1 ms to NHOH (Kohlmann and Poppe, 1999), which then reacts with OH or O2 to generate HNO:

$$NHOH + OH \longrightarrow HNO + H_2O$$
(R31)

$$NHOH + O_2 \longrightarrow HNO + HO_2$$
(R32)

280 The fate of HNO is the reaction with OH or  $O_2$  to generate NO (reactions R33 and R34):

 $HNO + O_2 \rightarrow NO + HO_2$  (R33)

$$HNO + OH \rightarrow NO + H_2O$$
 (R34)

High concentrations of  $O_3 \approx 10^{14}$  molecule cm-3) and HO2 ( $\approx 10^{11}$  molecule cm-3) in our system ensure that NO is converted to NO2 in less than 1s, explaining the non-observation of the IR absorption features of NO.

Finally, NO2 in this system will react with OH to form the main reactive nitrogen compound we observed, HNO3.

$$NO_2 + OH + M \rightarrow HNO_3 + M$$
 (R37)

Thus far we have not considered the reaction of the peroxy radical, OOCH2SO2NH2 with HO2 (Reaction R13), which is expected to result in the formation of a peroxide, HOOCH2SO2NH2. The most likely fate of HOOCH2SO2NH2 is reaction with OH for which (via comparison with CH3OOH) a rate coefficient close to  $1-5 \times 10^{-12}$  cm3 molecule-1 s-1 may be expected with H-abstraction from both the peroxide group (Reaction R38) or the adjacent carbon (Reaction R39).

$$HOOCH_2SO_2NH_2 + OH \rightarrow OOCH_2SO_2NH_2 + H_2O$$
(R38)

 $HOOCH_2SO_2NH_2 + OH \rightarrow HOOCHSO_2NH_2 + H_2O$ (R39)

295 Reaction (R38) regenerates the peroxy radical, whereas the HOOCHSO2NH2 radical may decompose (Reaction R40) to form formic acid HC(O)OH or via Reaction (R41) to form the same aldehyde that is generated in Reaction (R14), whilst regenerating OH:

$$HOOCHSO_2NH_2 \rightarrow HC(O)OH + SO_2NH_2$$
(R40)

$$HOOCHSO_2NH_2 \rightarrow OH + HC(O)SO_2NH_2$$
(R41)

300 The final products are thus the same as those resulting from the self-reaction of the peroxy radical. The path from MSAM to the observed end-products including the reactive intermediates that were not observed is illustrated in Fig. 7.

**3.4.2 Abstraction from the -NH2 group**

In analogy to the reaction between  $CH_3C(O)NH_2$  and OH (Barnes et al., 2010), H- abstraction from the - $NH_2$  group is expected to result in decomposition of the initially formed  $CH_3SO_2NH$  radical via C-S bond fission.

 $305 \quad CH_3SO_2NH \qquad \rightarrow \qquad CH_3 + SO_2NH \qquad (R42)$

The methyl radical would react with  $O_2$  to form the methyl-peroxy radical and in subsequent reactions (via CH3O) would result in CH2O formation. As discussed above CH2O will be efficiently oxidized to CO and CO2 in this system. However, the characteristic IR-absorption bands (Deng et al., 2016) of the SO2NH product were not observed in our experiments and calculations at the G4MP2 level of theory indicate that Reaction (R40) is endothermic (by 137 kJ mol-1). We conclude that

310 H-abstraction from the –NH2 group is a minor channel.

**3.5 Kinetic Simulation**

The proposed reaction mechanism (considering initial reaction by H-abstraction from the  $-CH_3$  group only) was tested by kinetic simulation using the KINTECUS program package (Ianni, 2015). The reactions used in the chemical scheme and the

associated rate coefficients are presented in Table S2. Where experimental rate coefficients were not available, we used rate

- 315 parameters from similar reactions, and rationalize these choices in the text associated with Table S2. Figure 8 shows the variation of the concentrations of the reagent, intermediates and products observed as a function of time in an experiment conducted at 298 K and 700 Torr of synthetic air. The error bars consider uncertainty associated with the absorption cross-sections (5 - 12%) and uncertainty in deriving the areas of the absorption bands areas (less than 3% in all cases). For MSAM, an uncertainty of ≈ 25% is expected, based on the indirect method of calibration (see Section 3.2). The
- 320 simulation results are depicted as solid lines. The good agreement with the N2O (formed from NH2 in Reaction R28) and HNO3 experimental data suggests that the fate of NH2 (the only source of reactive nitrogen in this system) is accurately described in the model. Note that the wall loss rate of HNO3 ( $1 \times 10^{-5}$  s-1) in the simulation was adjusted to match the HNO3 profile. The simulated amount of HNO3 lost to the wall at the end of the experiment was  $\approx 14\%$  of that formed, which helps to explain the non-unity yield of gas-phase nitrogen
- 325 compounds. The simulations indicate that the maximum concentrations of NO (7 × 109 molecule cm-3) and NO2 (~1012 molecule cm-3) are below the detection limit of the instrument, and explain why they were not observed. The strongest absorption features of HCHO (1700-1800 cm-1) overlap with those of H2O and HNO3 so that the predicted concentrations of HCHO (< 1012 molecule cm-3) are also below the detection limit.

The grey line in Fig. 8 represents the sum of  $SO_2 + SO_3 + H_2SO_4$ , i.e. all model trace gases containing sulphur, which, in the 330 absence of IR absorption features of  $SO_3$  or  $H_2SO_4$ , we equate to  $SO_2$ . We now draw attention to the fact that  $SO_2$  (the yield

of which is constant with time, see Fig. 6) is only well simulated if we neglect its removal by OH (Reaction R41).

$$OH + SO_2 + M \longrightarrow HOSO_2 + M$$
 (R43)

Otherwise, using the preferred rate constant (IUPAC, 2019) at 700 Torr and 298 K of  $9.0 \times 10^{-13}$  cm3 molecule-1 s-1 we find that the simulated SO2 concentration is significantly reduced and its yield is time dependent. At one bar of air, collisionally

335 stabilized HOSO2 is converted within 1  $\mu$ s to HO2 and SO3. In the atmosphere, SO3 reacts with H2O to form H2SO4 (R45). The conversion of SO3 to H2SO4 may be suppressed under our "dry" conditions.

SO2 should therefore not behave like a stable end-product in our experiments, but be converted to more oxidized forms. In order to confirm that SO2 is indeed stable in our experiments, we measured the relative rate of loss of SO2 and acetone under the same experimental conditions (Fig. S4 of the supplementary information). The apparent, relative rate constant  $k_{43} / k_5$  was 0.46, which converts to an effective rate constant for SO2 loss of 8.2 x 10-14 cm3 molecule-1 s-1. This is more than a factor of ten lower than the preferred value, indicating that the net rate of the OH-induced SO2 loss in our system is much lower than expected and not simply governed by the rate constant for the forward reaction to form HOSO2. The reformation of SO2

345 under our experimental conditions is subject of ongoing experiments in this laboratory, which are beyond the scope of the

present study. We note that the unexpected behaviour of SO2 does not significantly impact on the conclusions drawn in this work.

The simulation also captures the CO profile well, but fails to predict the early formation of CO2. The match between simulation and experiment could be improved to some extent for CO2 by amending the fate of the C(O)SO2NH2 radical as

350 described above (R17-R18) so that CO2 rather than CO is formed. The results (Fig S5 of the supplementary information) indicate that the improved simulation of CO2 is accompanied by complete loss of agreement with CO (which is no longer formed in measureable amounts) and poorer agreement with e.g. SO2 and HNO3. However, given that CO2 is generated from the cell walls during irradiation and cannot be used quantitatively (Section 3.5), the true fate of the  $C(O)SO_2NH_2$  radical remains obscure.

**355 **3.6 Atmospheric Implications**

370

The rate coefficient for a number of tropospheric, organo-sulphur trace gases are listed in Table 3. The rate coefficient for the title reaction is significantly lower than those for CH3SCH3 (CH3SCH3) and CH3S(O)CH3, (DMSO) for which reaction with OH is the major atmospheric loss process (lifetimes of hours), but comparable to CH3S(O)2CH3 which also has two S=O double bonds. However, as for most tropospheric trace gases, the lifetime of MSAM will be controlled by a number of

processes including photolysis, reactions with the three major oxidants, OH, NO3 and O3 as well as dry deposition ( $k_{dd}$ ) and 360 heterogeneous uptake to particles  $(k_{het})$ , followed by wet deposition.

The lack of C=C double-bonds in MSAM suggest that the reaction with  $O_3$  will be a negligible sink, which is confirmed by the low upper limit to the rate constant of  $1 \times 10^{-19}$  cm3 molecule-1 s-1 described in section 2.4. Whereas the reaction with NO3 represents an important loss mechanism for DMS, we do not expect this to be important for MSAM. CH3SCH3 reacts

365 with NO3 (despite lack of a C=C double bond) as the high-electron density around the sulphur atom enables a pre-reaction complex to form prior to H-abstraction. This mechanism is not available for MSAM because the electron density around the sulphur atom is reduced by the two oxygen atoms attached to it, which also provide steric-hindrance.

Owing to its low vapour pressure, we were unable to measure the UV-absorption spectrum of MSAM, but note that it was not photolysed at a measureable rate by the 254 nm radiation in our study. We conclude that photolysis in the troposphere, where actinic flux only at wavelengths above  $\geq 320$  nm is available is a negligible sink of MSAM.

Therefore, the lifetime of MSAM can be approximated by:

$$\tau_{\rm MSAM} = \frac{1}{k_4 [\rm OH] + k_{\rm dd} + k_{\rm het}} \tag{2}$$

Using our overall rate coefficient,  $k_4 = 1.4 \times 10^{-13} \text{ cm}^3$  molecule-1 s-1 for the title reaction and taking a diel-averaged OH concentration of 1 x 106 molecules cm-3, we can use equation (2) to calculate a first-order loss rate constant of  $k_4$ [OH] = 1.4  $\times 10^{-7}$  s-1. Which is equivalent to a lifetime of  $\approx 80$  days.

375

MSAM is highly soluble and a dry deposition velocity of  $\approx 1 \text{ cm s}^{-1}$  to the ocean has been estimated (Edtbauer et al., 2019). Combined with a marine boundary height of  $\approx 750 \pm 250$  m, this results in a loss rate coefficient of  $1.3 \times 10^{-5}$  s-1 or a lifetime with respect to uptake to the ocean of less than one day. Wet deposition is also likely to play a role, which may limit the MSAM lifetime to days under rainy conditions and to weeks in dry regions.

380 To a first approximation the heterogeneous loss rate of a trace gas to a particle is given by:

$$k_{het} = 0.25 \, \gamma \bar{c} A \tag{3}$$

where  $\gamma$  is the uptake coefficient which represents the net efficiency (on a per collision basis) of transfer of MSAM from the gas-phase to the particle phase),  $\bar{c}$  is the mean molecular velocity of MSAM (~26000 cm s-1) and A is the surface area density of particles (in cm2 cm-3) for which a typical value in low to moderately polluted regions would be 1 × 10-6 cm2 cm-3.

385 A rather low uptake coefficient of  $\sim 2 \times 10^{-5}$  would then be sufficient to compete with MSAM loss due to reaction with OH, but a value of  $2 \times 10^{-3}$  would be necessary to compete with dry-deposition.

**4** Conclusions**

The rate coefficient for reaction of methane sulphonamide (MSAM) with OH was determined using the relative rate method as  $(1.4 \pm 0.3) \times 10^{-13}$  cm3 molecule-1 s-1. The major, stable, quantifiable sulphur and nitrogen containing end-products of the reaction are SO2 and HNO3 with molar yields of (0.96 ± 0.15) and (0.62 ± 0.09), respectively. CO and CO2 are the dominant carbon-containing products. N2O 
[revised manuscript text omitted]

Hells, M. D. Valkowski, L. and Sahlagel, H. B.: Harmonia fragmency scaling factors for Hartras Fock, S. VWN, P. J. XP, P3.

Halls, M. D., Velkovski, J., and Schlegel, H. B.: Harmonic frequency scaling factors for Hartree-Fock, S-VWN, B-LYP, B3-LYP, B3-PW91 and MP2 with the Sadlej pVTZ electric property basis set., Theor Chem Acc., 105, 413–421, 2001.

450 IUPAC: Task Group on Atmospheric Chemical Kinetic Data Evaluation, (Ammann, M., Cox, R.A., Crowley, J.N., Herrmann, H., Jenkin, M.E., McNeill, V.F., Mellouki, A., Rossi, M. J., Troe, J. and Wallington, T. J.) http://iupac.pole-ether.fr/index.html., 2019.

Kohlmann, J.-P., and Poppe, D.: The tropospheric gas-phase degradation of  $NH_3$  and its impact on the formation of  $N_2O$  and  $NO_x$ , JAC, 32, 397-415, 1999.

- 455 Onel, L., Blitz, M., Dryden, M., Thonger, L., and Seakins, P.: Branching Ratios in Reactions of OH Radicals with Methylamine, Dimethylamine, and Ethylamine, Env. Sci. Tech., 48, 9935-9942, doi:10.1021/es502398r, 2014. Ray, A., Vassalli, I., Laverdet, G., and LeBras, G.: Kinetics of the thermal decomposition of the CH3SO2 radical and its reaction with NO2 at 1 torr and 298 K, J. Phys. Chem., 100, 8895-8900, doi:10.1021/jp9600120, 1996. Remko, M.: Theoretical study of molecular structure and gas-phase acidity of some biologically active sulfonamides, J.
- Phys. Chem. A, 107, 720-725, doi:10.1021/jp026980m, 2003.
  Spiro, P. A., Jacob, D. J., and Logan, J. A.: Global inventory of sulfur emissions with 1-degree x 1-degree resolution, J. Geophys. Res. -Atmos., 97, 6023-6036, doi:10.1029/91jd03139, 1992.

| Mode- | mode            |                         | Mode description    |                                   |       |                               |
|-------|-----------------|-------------------------|---------------------|-----------------------------------|-------|-------------------------------|
| symm. |                 | Experiment a | $6-31++(d,p)^{a,b}$ | Aug-CC-pVTZ a,b | ratio |                               |
| A''   | v 1  | 3476 (18.8)             | 3627 (18.8)         | 3612 (19.6)                       | 0.958 | NH 2 asym. stretch |
|       | v 3  |                         | 3193 (<0.1)         | 3169 (0.2)                        |       | CH 3 deformation   |
|       | ν 7  |                         | 1461 (1.1)          | 1457 (0.4)                        |       | CH 3 rocking       |
|       | $v_{10}$        | 1383 (100)              | 1322 (100)          | 1342 (100)                        | 1.048 | SO 2 asym. stretch |
|       | v 12 |                         | 1085 (2.2)          | 1087 (1.5)                        |       | NH 2 rocking       |
|       | $v_{14}$        |                         | 981 (0.3)           | 972 (0.3)                         |       | CH 3 twisting      |
|       | v 20 |                         | 385 (<0.1)          | 392 (<0.1)                        |       | C-S-N twist                   |
|       | v 21 |                         | 321 (1.1)           | 328 (1.2)                         |       | C-S-N twist                   |
|       | v 23 |                         | 218 (0.2)           | 216 (1.5)                         |       | CH 3 twist         |
|       | $v_{24}$        |                         | 170(14.9)           | 179(11.2)                         |       | NH 2 twist         |
| A'    | v 2  | 3380 (17.2)             | 3512 (13.0)         | 3508 (13.8)                       | 0.962 | NH 2 sym. stretch  |
|       | $\nu_4$         |                         | 3184 (<0.1)         | 3161 (<0.1)                       |       | CH 3 asym. stretch |
|       | ν 5  |                         | 3079 (0.1)          | 3065 (<0.1)                       |       | CH 3 sym. stretch  |
|       | $\nu_6$         | 1551 (15.6)             | 1591 (15.2)         | 1582 (13.5)                       | 0.975 | NH 2 bend          |
|       | ν 8  |                         | 1460 (2.5)          | 1456 (1.9)                        |       | CH 3 asym. bend    |
|       | v 9  | 1428 (3.7)              | 1363 (3.3)          | 1350 (4.2)                        | 1.048 | CH 3 umbrella      |
|       | v 11 | 1172 (72.8)             | 1115 (69.6)         | 1135 (64.2)                       | 1.051 | SO 2 sym. stretch  |
|       | v 13 | 976 (17.8)              | 994 (7.6)           | 987 (8.9)                         | 0.982 | CH 3 wagging       |
|       | v 15 | 857 (43.1)              | 867 (42.4)          | 864 (40.8)                        | 0.988 | NH 2 wagging       |
|       | $v_{16}$        |                         | 704 (7.2)           | 704 (4.2)                         |       | C-S stretch                   |
|       | v 17 |                         | 663 (81.9)          | 649 (81.5)                        |       | S-N stretch                   |
|       | v 18 |                         | 480 (14.9)          | 490 (15.8)                        |       | SO 2 wagging       |
|       | v 19 |                         | 457 (5.1)           | 468 (3.5)                         |       | SO 2 bend          |
|       | v 22 |                         | 285 (1.8)           | 290 (1.9)                         |       | C-S-N bend                    |

**Table 1.** Experimental and Calculated Vibrational Wavenumbers for MSAM.

a) Relative absorbance at band maximum in parentheses. b) Calculated using the B3LYP method.

| Table 2. Rate coefficients ratios and experimental condition | ns for the relative rat | te experiments. |
|--------------------------------------------------------------|-------------------------|-----------------|
|--------------------------------------------------------------|-------------------------|-----------------|

| Reference
reactant a | Concentration
(10 14 molecule cm -3 ) |                    |           |           | $\operatorname{Band}^{e}$ | k4 / kref         | $k_4$ (10 -13 cm 3 molecule -1 s -1 ) |                     |
|------------------------------------|----------------------------------------------------------------|--------------------|-----------|-----------|---------------------------|-------------------|-----------------------------------------------------------------------------------|---------------------|
| reactant                           | [MSAM] b                                            | [Ref] c | $[O_3]^d$ | $[H_2]^c$ | (em )                     |                   |                                                                                   | cule 5)             |
|                                    |                                                                |                    |           |           | 857                       | $0.792\pm0.012$   | $1.43\pm0.10$                                                                     |                     |
| Apatona                            | 0.21                                                           | 0.24               | 7.71      | 71.8      | 1172                      | $0.771 \pm 0.005$ | $1.39\pm0.09$                                                                     | $1.40 \pm 0.09^{f}$ |
| Acetone 0.31                       | 0.51                                                           | 0.34               |           |           | 1383                      | $0.779 \pm 0.006$ | $1.40\pm0.09$                                                                     |                     |
|                                    |                                                                |                    |           |           | 3380                      | $0.770\pm0.010$   | $1.39\pm0.10$                                                                     |                     |
| . .                         |                                                                |                    |           |           | 857                       | $0.312\pm0.004$   | $1.41\pm0.11$                                                                     |                     |
| Formic                             | 0.56                                                           | 0.55               | 5.74      | 65.2      | 1172                      | $0.311 \pm 0.002$ | $1.40\pm0.10$                                                                     | $1.38\pm0.09^{f}$   |
| Aciu                               |                                                                |                    |           |           | 1383                      | $0.302\pm0.002$   | $1.36\pm0.10$                                                                     |                     |
|                                    |                                                                |                    |           |           | 1172                      | $0.159 \pm 0.001$ | $1.43\pm0.17$                                                                     |                     |
| Methanol                           | 0.88                                                           | 0.25               | 4.58      | 46.6      | 1383                      | $0.153 \pm 0.001$ | $1.38\pm0.17$                                                                     | $1.42 \pm 0.16^{f}$ |
|                                    |                                                                |                    |           |           | 3380                      | $0.161 \pm 0.003$ | $1.45 \pm 0.19$                                                                   |                     |

475

a) Depletion of reference reactants monitored at 1221-1249, 1073-1133 and 2788-3070 cm-1 for acetone, formic acid and methanol, respectively.

b) Concentration estimated from the absorption cross section reported in Figure 1.

c) Concentration calculated from the measured pressures,

d) Concentration derived from the absorption cross section of O3

e) IR absorption bands of MSAM used for the determination of the concentration change over time,

f) Relative rate constant obtained from the linear fitting of all the data and using  $k_4 = (1.8 \pm 0.1) \times 10^{-13}$ ,  $k_5 = (4.5 \pm 0.36) \times 10^{-13}$ , and  $k_6 = (9.0 \pm 1.3) \times 10^{-13}$  cm3 molecule-1 s-1 rate constants respectively (IUPAC, 2019).

Table 3. Lifetimes of atmospheric organo-sulphur trace gases with respect to reaction with OH

|                                                 | k(OH) a    | Lifetime b | Reference                   |
|-------------------------------------------------|-----------------------|-----------------------|-----------------------------|
| CH 3 SO 2 NH 2 | $1.4 \times 10^{-13}$ | 80 days               | This work                   |
| CH 3 SO 2 CH 3 | $< 3 \times 10^{-13}$ | > 40 days             | (Falbe-Hansen et al., 2000) |
| CH 3 S(O)CH 3             | $5.9 \times 10^{-11}$ | 5 hours               | (Falbe-Hansen et al., 2000) |
| CH 3 SO 2 H               | 9.0×10 -11 | 2.8 hours             | (Burkholder et al., 2015)   |
| CH 3 SCH 3                | $2.2 \times 10^{-12}$ | 1.6 days              | (Atkinson et al., 2004)     |

a Units of cm3 molecule-1 s-1. bAssumes a diel average OH concentration of  $1 \times 10^6$  molecule cm-3.